# Oxaliplatin resistance in colorectal cancer enhances TRAIL sensitivity via death receptor 4 upregulation and lipid raft localization

**Joshua D Greenlee[1], Maria Lopez-Cavestany[1], Nerymar Ortiz-Otero[1], Kevin Liu[1], Tejas Subramanian[1], Burt Cagir[2], Michael R King[1]\***

[1]Vanderbilt University, Department of Biomedical Engineering PMB, Nashville, United States; [2]Donald Guthrie Foundation (DGF) for Research and Education Sayre, Sayre, United States

**Abstract** Colorectal cancer (CRC) remains a leading cause of cancer death, and its mortality is associated with metastasis and chemoresistance. We demonstrate that oxaliplatin-resistant CRC cells are sensitized to TRAIL-mediated apoptosis. Oxaliplatin-resistant cells exhibited transcriptional downregulation of caspase-10, but this had minimal effects on TRAIL sensitivity following CRISPR-Cas9 deletion of caspase-10 in parental cells. Sensitization effects in oxaliplatin-resistant cells were found to be a result of increased DR4, as well as significantly enhanced DR4 palmitoylation and translocation into lipid rafts. Raft perturbation via nystatin and resveratrol significantly altered DR4/raft colocalization and TRAIL sensitivity. Blood samples from metastatic CRC patients were treated with TRAIL liposomes, and a 57% reduction of viable circulating tumor cells (CTCs) was observed. Increased DR4/lipid raft colocalization in CTCs was found to correspond with increased oxaliplatin resistance and increased efficacy of TRAIL liposomes. To our knowledge, this is the first study to investigate the role of lipid rafts in primary CTCs.

*For correspondence:
mike.king@vanderbilt.edu

Competing interests: The authors declare that no competing interests exist.

## Introduction

Colorectal cancer (CRC) is the second leading cause of cancer death and is responsible for over 50,000 deaths annually in the United States (*Siegel et al., 2019*). The probability of being diagnosed with CRC in one's lifetime is 1 in 24, and there are over 100,000 new cases diagnosed annually in the United States alone. While the 5-year survival rate of localized and regional disease is 90 and 71%, respectively, patients with metastatic disease have just a 14% 5-year survival rate (*Early detection, diagnosis, and staging, 2021*). Dissemination to other organs is the cause of high mortality in most cancers as nearly 90% of all cancer deaths is attributed to metastasis (*Mehlen and Puisieux, 2006*). The most common sites of CRC metastases include the liver, lungs, and peritoneum (peritoneal carcinomatosis). While surgery and radiation remain curative options for patients with localized disease, the standard of care for CRC patients with advanced metastatic disease is commonly combination front-line chemotherapy treatment (*Werner and Heinemann, 2016*). These chemotherapy regimens typically include fluorouracil (5-FU) and leucovorin (LV) in combination, which work together to inhibit DNA and RNA synthesis and modulate tumor growth, extending median survival in patients from 9 months (with palliative care) to over 12 months (*Rodriguez-Bigas et al., 2003*). Oxaliplatin is a chemotherapeutic agent that upon binding to DNA forms DNA adducts to cause irreversible transcriptional errors, resulting in cellular apoptosis. When oxaliplatin is administered with 5-FU/LV (FOLFOX), the objective response rate is 50% in previously untreated patients, increasing the median overall survival to 18–24 months (*Rodriguez-Bigas et al., 2003*; *Briffa et al., 2017*).

While there have been incremental advances in extending survival using FOLFOX and other oxaliplatin-containing chemotherapeutics, patients who eventually succumb to the disease frequently develop chemoresistant subpopulations of cancer cells via intrinsic or acquired mechanisms (*Briffa et al., 2017*; *Martinez-Balibrea et al., 2015*). Mechanisms of oxaliplatin resistance in tumors include alterations in responses to DNA damage, cell death pathways (e.g., apoptosis, necrosis), NF-κB signaling, and cellular transport (*Martinez-Balibrea et al., 2015*). Despite the robustness of these oxaliplatin-resistant cancer cells, multiple studies suggest that chemoresistant subpopulations may be increasingly susceptible to adjuvant therapies (*Martinez-Balibrea et al., 2015*; *Sussman et al., 2007*; *Jeught et al., 2018*; *Combès et al., 2019*; *Ruiz de Porras et al., 2016*; *Cuello et al., 2001*). Tumor necrosis factor-related apoptosis-inducing ligand (TRAIL) is a member of the TNF family of proteins and induces apoptosis in cancer cells via binding to transmembrane death receptors (*von Karstedt et al., 2017*). The binding of TRAIL to trimerized death receptor 4 (DR4) and 5 (DR5) initiates an intracellular apoptotic cascade beginning with the recruitment of death domains and formation of the death-inducing signaling complex (DISC).

Lipid rafts (LRs) are microdomains in the plasma membrane lipid bilayer that are enriched in cholesterol and sphingolipids, with a propensity to assemble specific transmembrane and GPI-anchored proteins (*Simons and Toomre, 2000*). Mounting evidence has demonstrated that LRs play major roles in tumor progression, metastasis, and cell death (*Greenlee et al., 2021*). Studies have shown that translocation into LRs can augment signaling for a variety of cancer-implicated receptors, including growth factor receptors (IGFR and EGFR) and death receptors (Fas and Death receptors 4/5) (*Laurentiis et al., 2007*; *Marconi et al., 2013*; *Mollinedo and Gajate, 2020*; *George and Wu, 2012*). Translocation of death receptors into rafts enhances apoptotic signaling through the formation of clusters of apoptotic signaling molecule-enriched rafts (CASMER), which act as scaffolds to facilitate trimerization and supramolecular clustering of receptors (*Mollinedo and Gajate, 2020*). It has become increasingly evident that higher-order oligomerization of death receptors is necessary for effective apoptotic signaling in cancer cells (*Naval et al., 2019*).

Studies have shown that combination treatment of chemotherapeutic agents with TRAIL may sensitize cancer cells to TRAIL-mediated apoptosis through a variety of mechanisms, including death receptor upregulation (*Nagane et al., 2000*; *Gibson et al., 2000*; *Baritaki et al., 2007*), suppression of apoptotic inhibitors within the intrinsic pathway (*El Fajoui et al., 2011*), and redistribution of death receptors into LRs (*Xu et al., 2009*). However, no study has examined whether surviving oxaliplatin-resistant subpopulations of cancer cells have an enhanced sensitivity to TRAIL. In this study, we demonstrate that oxaliplatin-resistant cells show enhanced sensitivity to TRAIL-mediated apoptosis through LR translocation of DR4. Moreover, we elucidate mechanisms that drive this sensitization using chemoresistant cell lines and blood samples collected from metastatic cancer patients. The response of oxaliplatin-resistant CRC to TRAIL-based therapeutics may prove critical to establishing promising new adjuvants for patients who have exhausted conventional treatment modalities.

## Results

### Oxaliplatin-resistant cell lines show enhanced TRAIL sensitivity

Cell viability of four colorectal cancer cell lines after 24 hr treatment with 0.1–1000 ng/ml of TRAIL was measured and compared to the viability of oxaliplatin-resistant (OxR) cell lines (*Figure 1A*). Briefly, oxaliplatin-resistant cell lines were previously derived from exposure to increasing concentrations of oxaliplatin until a 10-fold increase in IC50 was achieved (*Dallas et al., 2009*; *Yang et al., 2006*; *Tanaka et al., 2015*). Parental and OxR cells were treated with a range of oxaliplatin concentrations to ensure that chemoresistance was conserved after multiple passages in culture (*Figure 1—figure supplement 1A*). Moreover, oxaliplatin-resistant cells were found to have increased invasion and motility compared to parental cells, consistent with literature reporting their derivation (*Figure 1—figure supplement 1B*). Interestingly, oxaliplatin-resistant HT29, SW620, and HCT116 cell lines showed increased maximum TRAIL sensitization levels compared to their parental counterparts, while SW480 cells showed similar or decreased sensitization levels (*Figure 1—figure supplement 2*). IC50 values demonstrate that a chemoresistant phenotype resulted in augmented TRAIL-mediated apoptosis in two cell lines (*Figure 1B*). Importantly, cells were not treated with any oxaliplatin in quantifying the level of TRAIL sensitization, and oxaliplatin was not supplemented in the cell culture

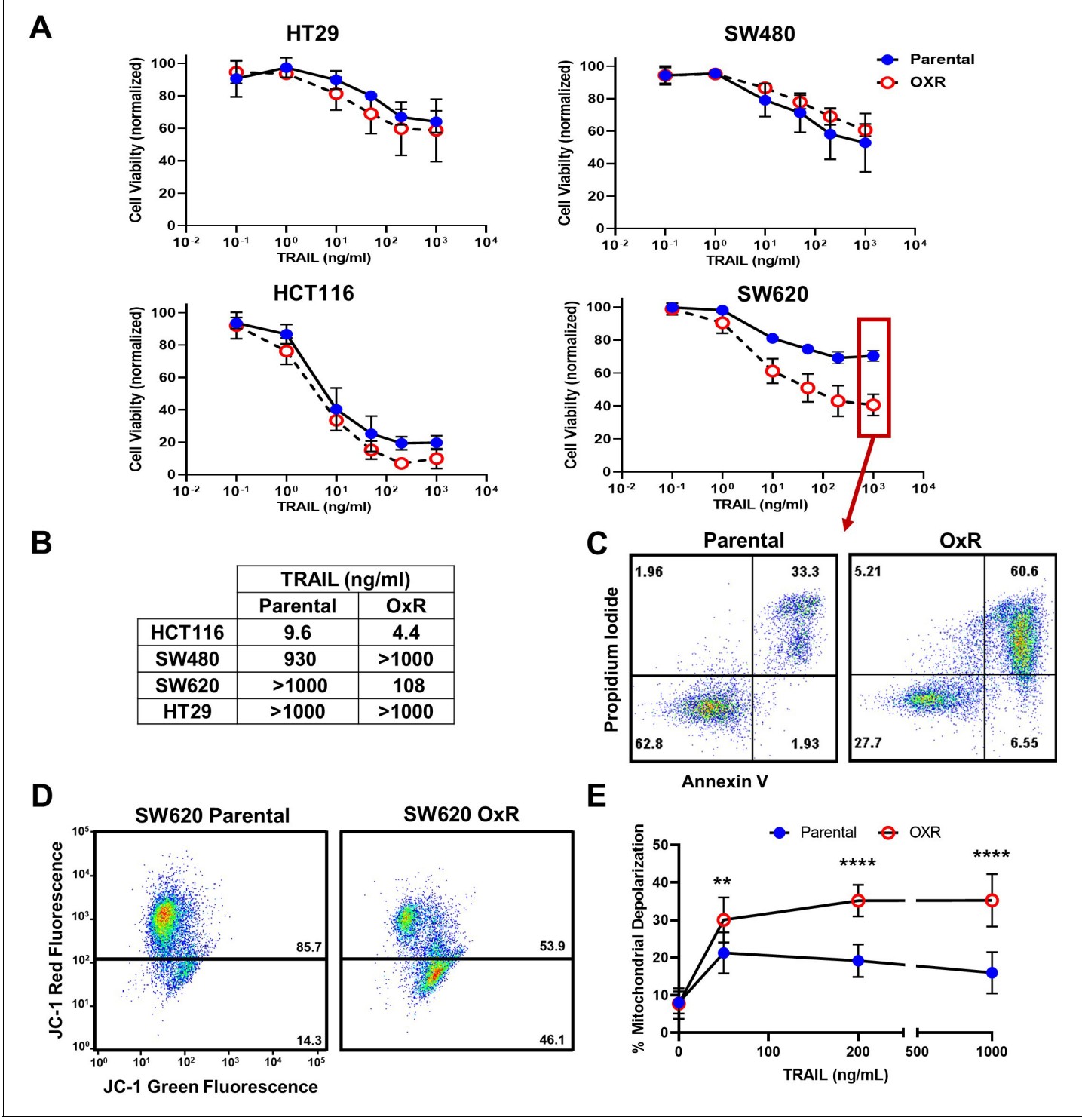

**Figure 1.** Oxaliplatin-resistant (OxR) colorectal cancer (CRC) cell lines exhibit enhanced sensitization to TRAIL-mediated apoptosis via the intrinsic pathway and mitochondrial permeabilization. (**A**) Oxaliplatin-resistant SW620, SW480, HCT116, and HT29 colon cancer cell lines demonstrate similar or enhanced sensitivity to TRAIL compared to their parental counterparts after 24 hr of treatment. N = 3 (biological replicates); n = 9 (technical replicates). (**B**) IC50 values were calculated using a variable slope four-parameter nonlinear regression. (**C**) Representative Annexin-V/PI flow plots comparing SW620 parental and OxR cell viability after 24 hr of treatment with 1000 ng/ml TRAIL. The four quadrants represent viable cells (bottom left), early apoptosis (bottom right), necrosis (top left), and late apoptosis (top right). (**D**) Representative flow plots of JC-1 assay after treatment with 1000 ng/ml of TRAIL. Mitochondrial depolarization is evidenced by decreased red fluorescence and increased green fluorescence. (**E**) Mitochondrial depolarization

*Figure 1 continued on next page*

*Figure 1 continued*

as a function of TRAIL concentration for SW620 parental and OxR cell lines. N = 3 (n = 9). For all graphs, data are presented as mean ± SD. **p<0.01; ****p<0.0001 (unpaired two-tailed t-test).

The online version of this article includes the following source data and figure supplement(s) for figure 1:

**Source data 1.** Raw viable cell counts from Annexin-V/PI assays and percent depolarized mitochondria in SW620 cells (panels A and E).
**Figure supplement 1.** Oxaliplatin-resistant colorectal cancer (CRC) cells retain their increasingly chemoresistant and invasive phenotypes in culture.
**Figure supplement 1—source data 1.** Raw data from MTT cell viability assays after oxaliplatin treatment (panel A).
**Figure supplement 1—source data 2.** Invasive cell counts from Transwell assays (panel B).
**Figure supplement 2.** Sensitization of oxaliplatin-resistant colorectal cancer (CRC) cell lines to TRAIL.
**Figure supplement 2—source data 1.** TRAIL sensitization calculations.
**Figure supplement 3.** HCT116 oxaliplatin-resistant (OxR) cells have increased mitochondrial depolarization and activation of the intrinsic apoptotic pathway when treated with TRAIL.
**Figure supplement 3—source data 1.** Percent depolarized mitochondria in HCT116 cells measured from JC-1 assays.

media to exclude any possible effects from combination treatment. In SW620 cells, large differences in apoptosis were observed when treated with the highest concentration of TRAIL (1000 ng/ml) (*Figure 1C*). Only 33.3% of parental cells were found to be in late-stage apoptosis after 24 hr compared to 60.6% for OxR cells.

To determine if the observed differences in apoptosis were due to enhanced mitochondrial outer membrane permeability, a JC-1 dye was used. SW620 OxR cells exhibited over a threefold increase in the population of JC-1 red (-) cells, indicating increased mitochondrial depolarization (*Figure 1D*). Mitochondrial depolarization was significantly enhanced in OxR cells for TRAIL concentrations of 50 ng/ml and higher (*Figure 1E*). Similar TRAIL-induced mitochondrial effects were observed in HCT116 cells (*Figure 1—figure supplement 3*). These results demonstrate that enhanced TRAIL-mediated apoptosis is occurring, at least in part, via the intrinsic pathway and mitochondrial disruption.

## Oxaliplatin-resistant derivatives have decreased CASP10 that has little consequence on TRAIL sensitization

Given that enhanced TRAIL-mediated apoptosis was found to occur via the mitochondrial pathway, gene expression of apoptotic transcripts was compared between the parental and OxR cells. RT-PCR human apoptosis profiler arrays were used to analyze transcripts within the SW620 and HCT116 cell lines since these cells showed the highest degree of OxR TRAIL sensitization and exhibit different innate sensitivities to TRAIL (HCT116 cells are TRAIL-sensitive, whereas SW620 cells are TRAIL-resistant). Interestingly, upon analyzing the RNA expression of 84 apoptotic transcripts, both cell lines shared similar profiles between parental and OxR derivatives. HCT116 OxR cells showed upregulated pro-apoptotic transcripts cytochrome-c and caspase-4 (*Figure 2A*). Cytochrome-c is released from the mitochondria into the cytosol after mitochondrial permeabilization, binding to adaptor molecule apoptosis-protease activating factor 1 (Apaf-1) to form the apoptosome and initiate downstream caspase signaling (*Garrido et al., 2006*). Caspase-4 is localized to the ER and initiates apoptosis in response to ER stress (*Hitomi et al., 2004*). Interestingly, SW620 OxR cells had upregulated Fas, a cell surface death receptor that acts similarly in apoptotic signaling to DR4/DR5 via binding of Fas ligand (*Özören and El-Deiry, 2003*), and osteoprotegerin, a soluble decoy receptor that sequesters TRAIL and inhibits apoptosis (*Sandra et al., 2006*; *Figure 2B*). Upregulated Fas expression in SW620 OxR cells was confirmed via surface staining and flow cytometry; however, receptor neutralization with the ZB4 anti-Fas antibody had no effect on TRAIL sensitization when treated in combination with TRAIL (*Figure 2—figure supplement 1A–C*). Notably, both HCT116 and SW620 OxR cell lines had caspase-10 as the most significantly downregulated transcript. To determine whether this was of consequence to the observed TRAIL sensitization, an SW620 caspase-10 knockout cell line was created using a multi-guide sgRNA CRISPR-Cas9 approach. Knock-out (KO) efficiency was found to be 93% (*Figure 2C*). The TRAIL sensitivity of this caspase-10 KO cell line was compared to a control cell line treated with Cas9 only. Caspase-10 KO cells showed only a slight decrease in cell viability after 24 hr of TRAIL treatment (*Figure 2D*). The number of late-stage apoptotic cells remained similar between cell lines (*Figure 2E*), and the maximum TRAIL sensitization observed was insignificant following caspase-10 KO (*Figure 2F*).

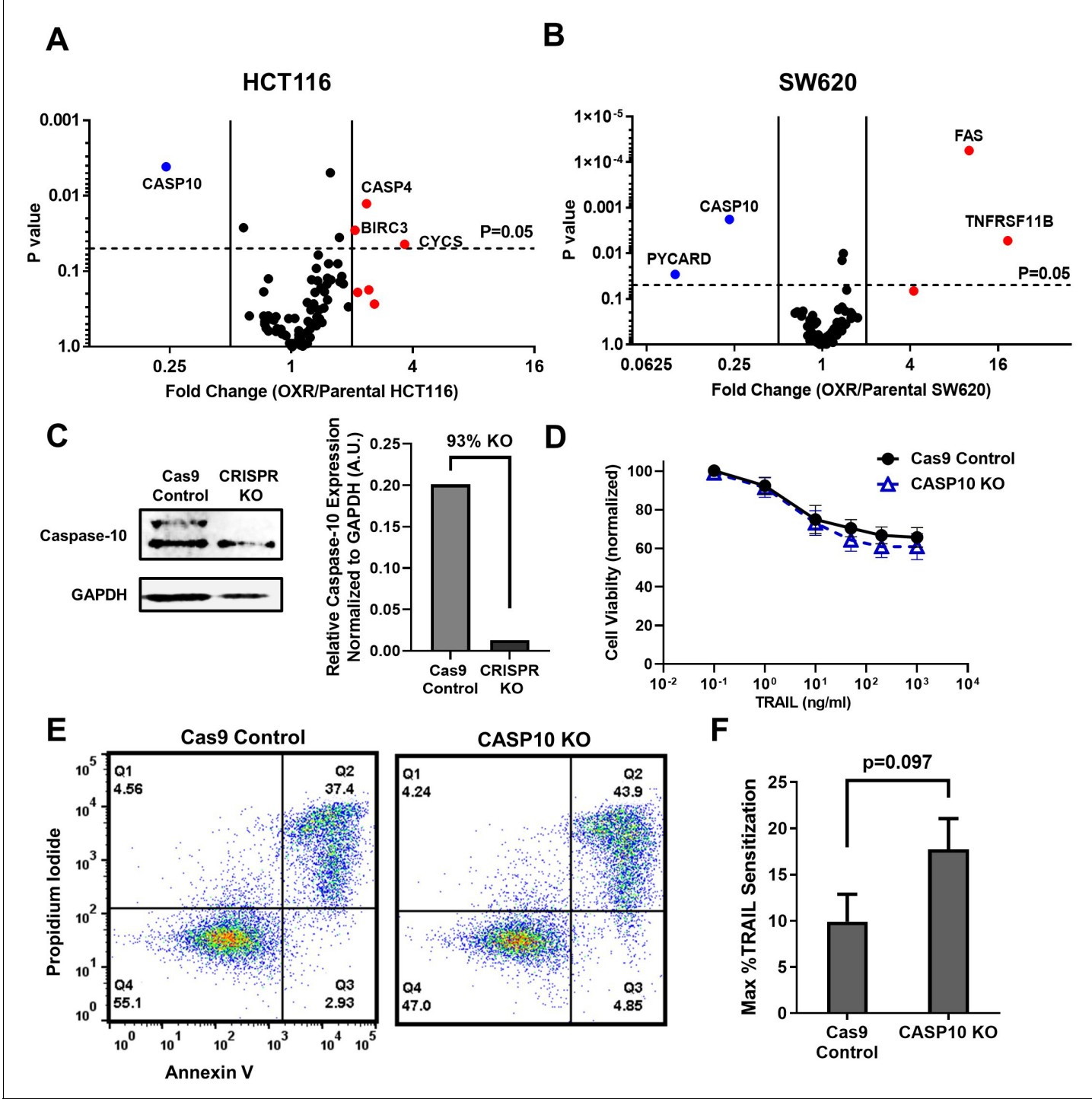

**Figure 2.** Microarray profiles show that parental and oxaliplatin-resistant (OxR) colorectal cancer (CRC) cell lines have similar expression of apoptotic transcripts while OxR derivatives have significantly downregulated CASP10. (A, B) Volcano plots of RT-PCR Apoptosis Profiler arrays demonstrate downregulation of CASP10 in OxR phenotypes. N = 3. (C) CRISPR/Cas9 knockout of caspase-10 in SW620 parental cells was confirmed via western blot. sgRNA/Cas9 ribonucleoprotein complexes reduced caspase-10 expression by 93% compared to cells treated with Cas9 alone. (D) CASP10 knock-out (KO) cells demonstrate slight decreases in viability when treated with TRAIL compared to Cas9 control. Data are presented as mean ± SD. N = 3 (n = 9). (E) Representative Annexin-V/PI flow plots comparing SW620 parental (Cas9 only) and CASP10 KO cell viability after 24 hr of treatment with 1000 ng/ml TRAIL. (F) Depletion of caspase-10 did not have a significant effect on TRAIL sensitization (unpaired two-tailed t-test). Data are presented as mean + SEM. N = 3 (n = 9).

The online version of this article includes the following source data and figure supplement(s) for figure 2:

*Figure 2 continued on next page*

*Figure 2 continued*

**Source data 1.** Apoptosis microarray data in HCT116 cells with fold regulation calculations generated using the GeneGlobe Data Analysis Center (panel A).

**Source data 2.** Apoptosis microarray data in SW620 cells with fold regulation calculations generated using the GeneGlobe Data Analysis Center (panel B).

**Source data 3.** Western blot images (raw and annotated) confirming CASP10 KO (panel C).

**Source data 4.** Quantification of CASP10 KO from western blots (panel C).

**Source data 5.** Cell viability and TRAIL sensitization calculations in CASP10 KO cells (panels D and F).

**Figure supplement 1.** Upregulated FasR in SW620 oxaliplatin-resistant (OxR) cells has no effect on TRAIL sensitivity.

**Figure supplement 1—source data 1.** Cell apoptosis and TRAIL sensitization calculations after ZB4 treatment with TRAIL.

## TRAIL-sensitized OxR cell lines have upregulated DR4

While changes in death receptor expression were insignificant at a transcriptional level, studies have demonstrated that chemoresistance can alter receptor abundance via mechanisms of translational regulation (*Si et al., 2019*; *Just et al., 2019*). Confocal microscopy showed that oxaliplatin-resistant cells have increased DR4 in both HCT116 (*Figure 3A*) and SW620 (*Figure 3B*) cell lines. To quantify receptor expression, total DR4 area per cell was analyzed for at least 70 cells. Analysis showed OxR derivative cell lines had significantly increased DR4 area per cell (*Figure 3C*). There were no differences in cell size between parental and OxR derivatives for all four cell lines (*Figure 3—figure supplement 1*). Flow cytometry staining of non-permeabilized cells was used to determine if this death receptor increase was also observed on the cell surface. Both HCT116 OxR and SW620 OxR cells showed significant increases in DR4 surface expression (*Figure 3D*). Total and surface DR4 expression was similar between parental and OxR derivatives in mildly sensitized HT29 cells and unsensitized SW480 cells (*Figure 3—figure supplement 2A–C*). To account for possible thresholding effects in area quantification, raw integrated density counts per cell were also measured and found to be consistent with changes in receptor area (*Figure 3—figure supplement 3*). Increases in DR4 expression of OxR derivatives were also confirmed via western blot but were only significant in SW620 cells (*Figure 3E, F*). OxR HCT116, SW620, and HT29 cells all displayed increases in DR5 area per cell, while SW480 OxR cells had significant decreases in total DR5 expression (*Figure 3—figure supplement 4A–D*). However, total receptor area per cell was considerably lower for DR5 compared to DR4. Additionally, expression of surface DR5, analyzed via flow cytometry, was only significantly upregulated in SW620 OxR cells (*Figure 3—figure supplement 4E*). This is further confounded by western blot data, which show no change in DR5 expression in HCT116 OxR cells, and a significant decrease in SW620 OxR cells (*Figure 3—figure supplement 5A, B*). Decoy receptors are surface receptors that, like death receptors, can bind to exogenous TRAIL. However, decoy receptor 1 (DcR1) and decoy receptor 2 (DcR2) are unable to activate the apoptotic pathway, making these receptors sequestering agents that competitively bind to TRAIL. While some studies have shown that chemotherapy-induced changes in TRAIL sensitivity have been linked to modulation or augmentation of decoy receptors (*Toscano et al., 2008*), all cell lines exhibited no meaningful difference in surface DcR1 and DcR2 expression between parental and OxR derivatives (*Figure 3—figure supplement 6*). Despite statistical significance in SW480 and HT29 cells, decoy receptor expression, especially DcR2, was expressed in negligible quantities in these cell lines.

Given the consistency in data suggesting DR4 upregulation in TRAIL-sensitized OxR cell lines, cells were treated with the DR4-agonist monoclonal antibody mapatumumab to determine DR4 specificity. SW620 OxR cells exhibited significant increases in the number of apoptotic cells after 24 hr of treatment, including a threefold increase in apoptosis at concentrations of 10 µg/ml (*Figure 3G*). Cell viability closely paralleled TRAIL treatments: parental cells remained resistant at high doses while OxR cells exhibited a dose-responsive decrease in cell viability (*Figure 3H*). HCT116 cell lines exhibited similar results as OxR cells were significantly more apoptotic at concentrations exceeding 0.1 µg/ml (*Figure 3—figure supplement 7A, B*). The maximum mapatumumab sensitization was calculated to be greater than 40% for oxaliplatin-resistant HCT116 and SW620 cells (*Figure 3I*), providing more causal evidence for a DR4-associated mechanism.

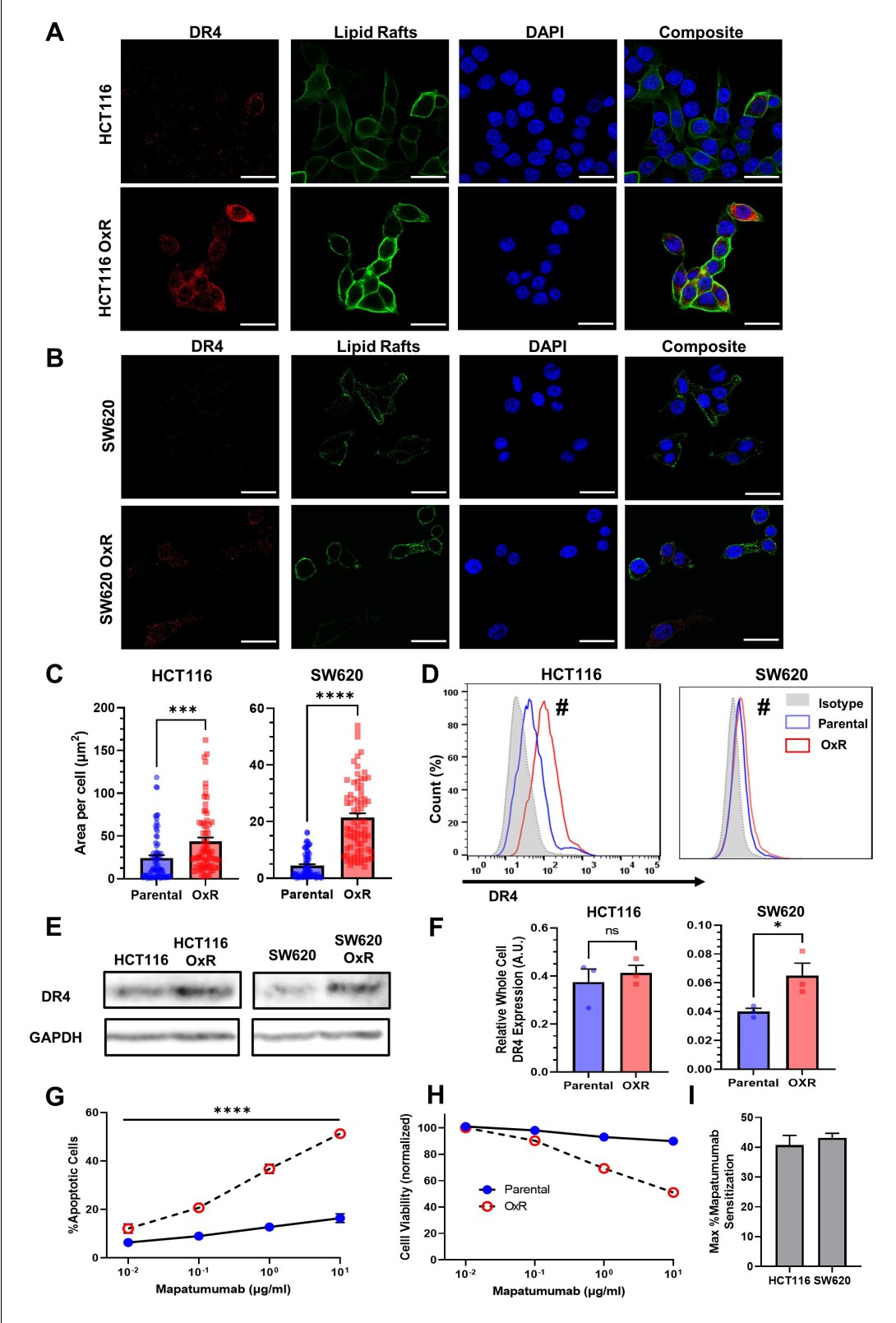

**Figure 3.** Oxaliplatin-resistant (OxR) colon cancer cell lines have upregulated DR4 expression. (A, B) Confocal micrographs of HCT116 and SW620 cells, respectively. Red channel represents DR4, green is lipid rafts, and blue is DAPI (nuclei). Scale bar = 30 μm. (C) Quantification of DR4 area per cell in HCT116 and SW620 cells. For each cell line, N = 75 cells were analyzed. Data are presented as mean + SEM from N = 3 independent experiments. ***p<0.001; ****p<0.0001 (unpaired two-tailed t-test). (D) OxR cells had increased surface expression of DR4 in non-permeabilized cells analyzed via

*Figure 3 continued on next page*

*Figure 3 continued*

flow cytometry. #Significant according to a chi-squared test (see *Supplementary file 1*). (**E**) Western blots for DR4 in whole cell lysates of parental and OxR cells. (**F**) Quantification of western blots from three independent experiments (N = 3). Data are presented as mean + SEM. *p<0.05 (unpaired two-tailed t-test). (**G**) Percentage of apoptotic SW620 cells after treatment with 0.01–10 µg/ml mapatumumab (sum of early and late-stage apoptotic cells from Annexin/PI staining). Data are presented as mean ± SD. N = 3 (n = 6). ****p<0.0001 (multiple unpaired two-tailed t-tests). (**H**) Cell viability of SW620 cells after mapatumumab treatment, determined by Annexin-V/PI staining. Data are presented as mean ± SD. N = 3 (n = 6). (**I**) Maximum mapatumumab sensitization within OxR cell lines compared to their parental counterparts. Data are presented as mean + SEM.

The online version of this article includes the following source data and figure supplement(s) for figure 3:

**Source data 1.** Quantification of DR4 area per cell in HCT116 and SW620 cells (panel C).

**Source data 2.** Western blot images (raw and annotated) for DR4 (panel E).

**Source data 3.** Quantification of DR4 from western blots (panel F).

**Source data 4.** Cell viabilty and percent apoptosis in SW620 cells after mapatumumab treatment (panels G-I).

**Figure supplement 1.** Parental and oxaliplatin-resistant (OxR) cell lines have no significant changes in cell area, analyzed from confocal microscopy images.

**Figure supplement 1—source data 1.** Cell area measurements for all cell lines.

**Figure supplement 2.** Unsensitized SW480 and HT29 oxaliplatin-resistant (OxR) cell lines show no significant changes in DR4 expression or lipid raft colocalization relative to their parental counterparts.

**Figure supplement 2—source data 1.** Quantification of DR4 and lipid raft colocalized DR4 area per cell in SW480 and HT29 cells.

**Figure supplement 3.** Sensitized oxaliplatin-resistant cell lines have significantly increased DR4 integrated density per cell.

**Figure supplement 3—source data 1.** Integrated density measurements of DR4.

**Figure supplement 4.** Chemoresistant HCT116, SW620, and HT29 cells have upregulated DR5 while in chemoresistant SW480 cells, DR5 is decreased.

**Figure supplement 4—source data 1.** Quantification of DR5 area per cell for all cell lines.

**Figure supplement 5.** Western blots show TRAIL-sensitized HCT116 (A) and SW620 (B) oxaliplatin-resistant (OxR) cells have no increases in DR5.

**Figure supplement 5—source data 1.** Quantification of DR5 from western blots.

**Figure supplement 5—source data 2.** Western blot images (raw and annotated) for DR5.

**Figure supplement 6.** Flow cytometry analysis of the surface expression of decoy death receptors 1 (DcR1) and 2 (DcR2) on nonpermeabilized parental and oxaliplatin-resistant (OxR) cell lines.

**Figure supplement 7.** HCT116 oxaliplatin-resistant (OxR) cells are increasingly sensitive to DR4 agonist antibody treatment.

**Figure supplement 7—source data 1.** Cell viabilty and percent apoptosis in HCT116 cells after mapatumumab treatment.

## TRAIL-sensitized OxR cell lines have enhanced colocalization of DR4 into LRs

Binary projections of colocalization events between DR4 and LRs demonstrate that OxR phenotypes had enhanced DR4 translocation into LRs in HCT116 and SW620 cells (*Figure 4A*). Quantification of total area of colocalization events showed that HCT116 OxR and SW620 OxR cells have significantly enhanced DR4 localized into LRs, each with an over fourfold increase (*Figure 4B*). The areas of DR4/LR colocalized events per cell were not significantly different in HT29 and SW480 cells (*Figure 3— figure supplement 2D*). Other methods of colocalization analysis, including calculation of the Manders' Correlation Coefficient (MCC), supported these results, specifically in HCT116 and SW620 cell lines where the Manders' overlap was significantly greater in OxR cells (*Figure 4—figure supplement 1*). The fold change in DR4/LR colocalization area between OxR and parental cells exhibited a strong linear correlation (0.86) with TRAIL sensitization (*Figure 4C*). Colocalization of LRs with DR5 was significantly enhanced only in HCT116 and HT29 cells, and analysis of the correlation between DR5 colocalization and TRAIL sensitization resulted in a weaker correlation of 0.48 (*Figure 4—figure supplement 2A–C*). Quantification of LR area per cell revealed insignificant changes between parental and OxR derivatives in all cell lines except for HT29 cells, where parental cells showed significantly more rafts (*Figure 4—figure supplement 3*).

To confirm DR4 redistribution into rafts, western blots for DR4 were run on plasma membrane-derived LR fractions, isolated using non-ionic detergent and centrifugation. Isolated LR fractions exhibited significant increases in DR4 for both HCT116 OxR and SW620 OxR cells (*Figure 4D, E*). β-Actin was used as a loading control to compare relative DR4 expression between parental and OxR cell lines (*Suprynowicz et al., 2008*). There were no detectable levels of DR5 in western blots from LR isolated fractions (*Figure 4—figure supplement 2D*). This is consistent with studies in hematological cancers that demonstrate raft localization of DR4 but not DR5 (*Marconi et al., 2013*; *Xiao et al., 2011*). To further examine the proximity of DR4 and LRs between phenotypes, Förster resonance energy transfer (FRET) efficiency was measured using a previously described flow

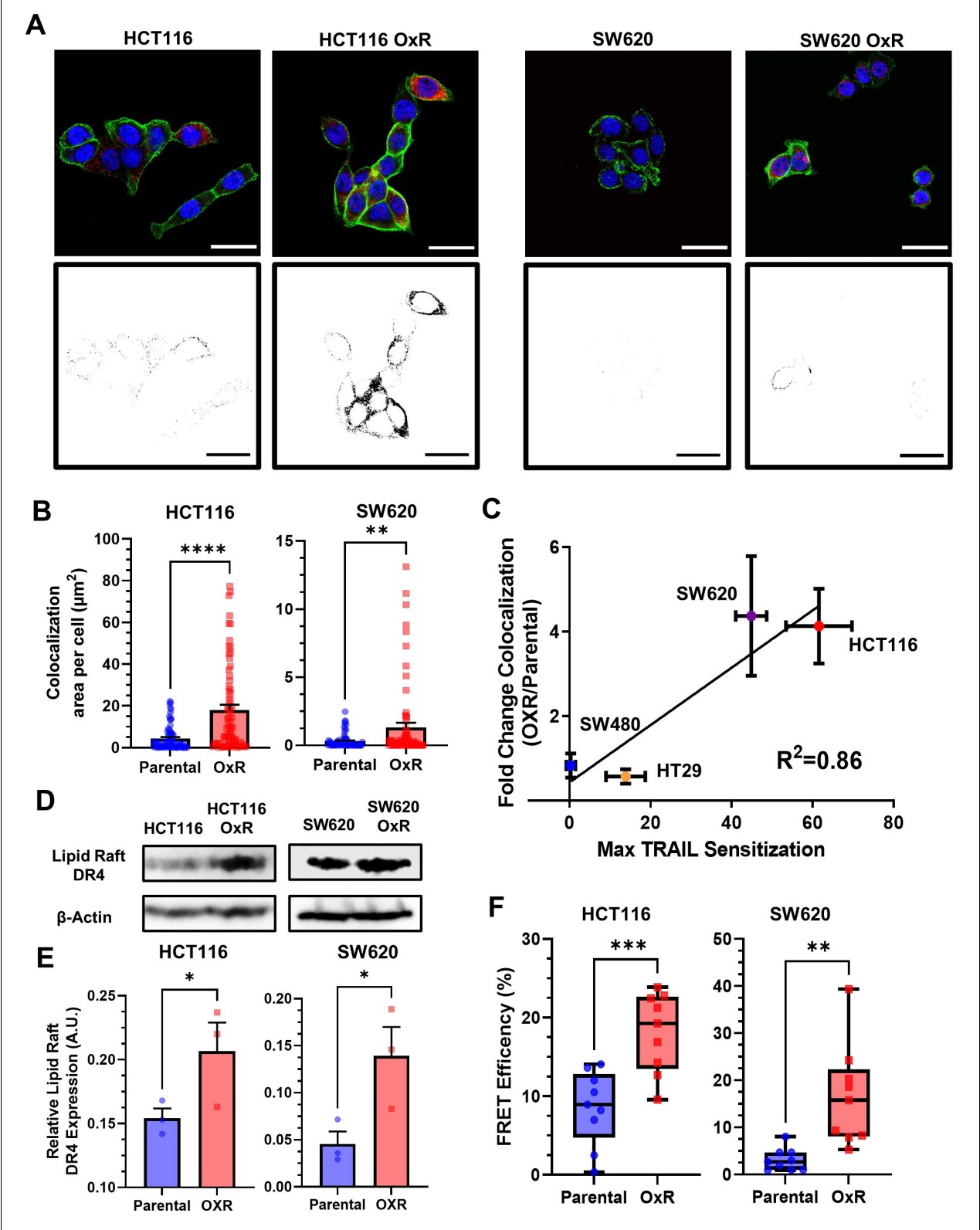

**Figure 4.** Oxaliplatin-resistant (OxR) colon cancer cell lines have enhanced colocalization of DR4 into lipid rafts. (**A**) Composite images and binary projections of DR4/LR colocalization areas in HCT116 and SW620 cell lines. Lipid raft and DR4 binary images were generated for a specified threshold, then multiplied by one another to generate images with positive pixels in double-positive areas. Red is DR4, green is lipid rafts, and blue is DAPI. Scale bar = 30 µm. (**B**) Quantification of DR4 and lipid raft colocalization area per cell in HCT116 and SW620 cells. For each cell line, N = 75 cells were

*Figure 4 continued on next page*

*Figure 4 continued*

analyzed. **p<0.01; ****p<0.0001 (unpaired two-tailed t-test). (C) Correlation between the fold change in DR4/LR colocalization (OxR phenotype/parental) and maximum TRAIL sensitization observed by the OxR phenotype for each of the four cell lines (simple linear regression analysis). (D) Lipid raft fractions were isolated and analyzed for DR4 via western blot in parental and OxR cells. (E) Quantification of lipid raft DR4 blots in (D). *p<0.05 (unpaired two-tailed t-test). For all graphs, data are presented as mean + SEM. (F) Förster resonance energy transfer (FRET) efficiencies of FITC-labeled DR4 (donor) and Alexa 555-labeled lipid rafts (acceptor) in parental and OxR cells analyzed via flow cytometry. **p<0.01; ***p<0.001 (unpaired two-tailed t-test).

The online version of this article includes the following source data and figure supplement(s) for figure 4:

**Source data 1.** Quantification of LR-colocalized DR4 area per cell in HCT116 and SW620 cells (panel B).
**Source data 2.** Correlation analysis of LR-colocalized DR4 area per cell with TRAIL sensitization (panel C).
**Source data 3.** Western blot images (raw and annotated) for DR4 from LR isolates (panel D).
**Source data 4.** Quantification of LR DR4 from western blots (panel E).
**Source data 5.** FRET efficiency calculations (panel F).
**Figure supplement 1.** TRAIL-sensitized oxaliplatin-resistant (OxR) cells have significantly increased colocalization of DR4 with lipid rafts according to Manders' Correlation Coefficient.
**Figure supplement 1—source data 1.** MCC calculations for all cell lines.
**Figure supplement 2.** Sensitization to TRAIL in oxaliplatin-resistant (OxR) cell lines poorly correlates with DR5 expression while lipid raft fractions have no detectable DR5.
**Figure supplement 2—source data 1.** Quantification of LR-colocalized DR5 area per cell (panel A).
**Figure supplement 2—source data 2.** Correlation analysis of DR5 and LR-colocalized DR5 area per cell with TRAIL sensitization (panels B and C).
**Figure supplement 2—source data 3.** Western blot images (raw and annotated) for DR5 from LR isolates (panel D).
**Figure supplement 3.** Quantification of lipid raft area per cell.
**Figure supplement 3—source data 1.** Quantification of LR area per cell.

cytometry protocol and calculated via a donor quenching method (*Ujlaky-Nagy et al., 2018*). Both HCT116 and SW620 OxR cells had significantly increased FRET efficiencies compared to their parental counterparts, with an over twofold and fivefold increase, respectively (*Figure 4F*).

## Altering LR composition affects DR4/LR colocalization and has consequential effects on TRAIL sensitization

To probe the effects of LR modulation on DR4 clustering and TRAIL sensitization, SW620 OxR and HCT116 OxR cells were treated with 5 µM of nystatin, a cholesterol-sequestering agent that inhibits LR formation, in combination with TRAIL for 24 hr (*Figure 5A, G*). Nystatin inhibited TRAIL-mediated apoptosis in SW620 OxR cells, significantly decreasing the maximum TRAIL sensitization from 45% to 23% (*Figure 5B*). Nystatin treatment was found to decrease DR4/LR colocalization by over 20-fold (*Figure 5C, M*). Similar results were found in HCT116 OxR cells as nystatin treatment decreased TRAIL sensitization from 62% to 1% (*Figure 5H*) and decreased DR4/LR colocalization by over ninefold (*Figure 5I*). To demonstrate that enhancing LR formation would have pro-apoptotic effects, parental cells were treated with 70 µM of resveratrol in combination with TRAIL for 24 hr (*Figure 5D, J*). Resveratrol has been shown to stabilize liquid-ordered domains in the plasma membrane and promote cholesterol/sphingolipid-enriched LRs (*Neves et al., 2016*). Resveratrol significantly sensitized parental SW620 cells to TRAIL irrespective of TRAIL concentration with a maximum TRAIL sensitization of 68% (*Figure 5E*). Treatment with resveratrol coincided with significant augmentation of DR4/LR colocalization area, an increase of over sixfold (*Figure 5F, N*). Similarly, parental HCT116 cells treated with resveratrol were sensitized 59% (*Figure 5K*), corresponding with a nearly sevenfold increase in DR4/LR colocalization area per cell (*Figure 5L*). Resveratrol and nystatin had no significant effects on DR5 LR colocalization, except in SW620 OxR cells where nystatin treatment surprisingly resulted in a slight increase in colocalization (*Figure 5—figure supplement 1A, B*).

## S-Palmitoylation of DR4 is enhanced in OxR cells

Palmitoylation is the reversible, post-translational addition of the saturated fatty acid palmitate to the cystine residue of proteins. Palmitoylation of DR4 has proven to be critical for receptor oligomerization and LR translocation, both obligatory for effective TRAIL-mediated apoptotic signaling (*Rossin et al., 2009*). S-Palmitoylation of DR4 in SW620 parental and OxR cells was analyzed via protein precipitation, free thiol blocking, thioester cleavage of palmitate linkages, and exchange with a

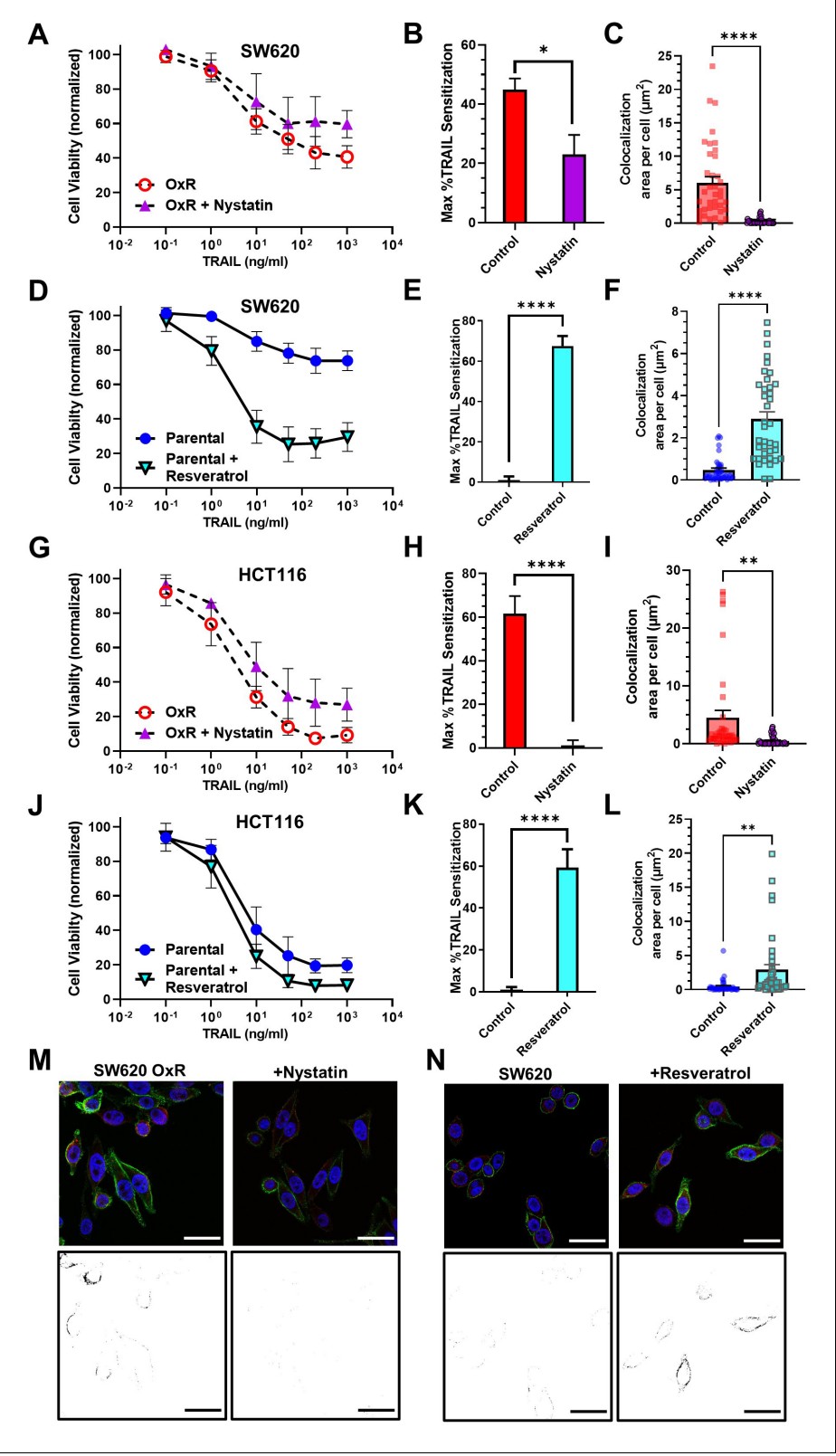

**Figure 5.** Pharmacological perturbation of DR4 localization in lipid rafts significantly alters cellular apoptosis in response to TRAIL. (**A, G**) SW620 oxaliplatin-resistant (OxR) and HCT116 OxR cells, respectively, treated for 24 hr with a combination of TRAIL and 5 μM nystatin. (**B, H**) SW620 OxR and HCT116 OxR cells, respectively, showed a significant decrease in TRAIL sensitization when treated in combination with nystatin. N = 3 (n = 9). (**C, I**) Treatment with 5 μM nystatin significantly decreased DR4/LR colocalization area in SW620 OxR and HCT116 OxR cells, respectively. For each cell line,

*Figure 5 continued on next page*

*Figure 5 continued*

N = 40 cells were analyzed. (**D, J**) SW620 Par and HCT116 Par cells, respectively, treated for 24 hr with a combination of TRAIL and 70 µM resveratrol. N = 3 (n = 9). (**E, K**) SW620 Par and HCT116 Par cells, respectively, showed a significant increase in TRAIL sensitization when treated in combination with resveratrol. N = 3 (n = 9). (**F, L**) Treatment with 70 µM nystatin significantly increased DR4/LR colocalization area in SW620 Par and HCT116 Par cells, respectively. For each cell line, N = 40 cells were analyzed. (**M**) Representative composite images and binary projections of DR4/LR colocalization in SW620 OxR cells before and after nystatin treatment. (**N**) Representative composite images and binary projections of DR4/LR colocalization in parental SW620 cells before and after resveratrol treatment. Red represents DR4, green is lipid rafts, and blue is DAPI. Scale bar = 30 µm. **p<0.01; ****p<0.0001 (unpaired two-tailed t-test for all graphs). (**A, D, G, J**) Data are presented as mean ± SD. (**B, C, E, F, H, I, K, L**) Data are presented as mean + SEM.

The online version of this article includes the following source data and figure supplement(s) for figure 5:

**Source data 1.** Cell viability after TRAIL combination treatments with resveratrol and nystatin (panels A, D, G, J).

**Source data 2.** TRAIL sensitization calculations after resveratrol and nystatin (panels B, E, H, K).

**Source data 3.** Quantification of LR-colocalized DR4 after resveratrol and nystatin treatment (panels C, F, I, L).

**Figure supplement 1.** Quantification of the effects of resveratrol and nystatin on DR5 colocalization with lipid rafts in HCT116 (**A**) and SW620 cells (**B**).

**Figure supplement 1—source data 1.** Quantification of the effects of resveratrol and nystatin on DR5 colocalization with LRs in HCT116 and SW620 cells.

mass tag label to quantify the degree of palmitoylated protein. We discovered that DR4 has four distinct palmitoylated sites, the degree of which was enhanced in the oxaliplatin-resistant phenotype (*Figure 6A*). Quantifying the percentage of palmitoylated protein in relation to input fraction (IFC) and non-mass tag preserved controls (APC-) validated that OxR cells had a significantly higher percentage of DR4 that was palmitoylated (55% compared to 43%) (*Figure 6B*). To determine whether enhanced palmitoylation was specific to DR4 and not a ubiquitous characteristic of the OxR phenotype, total cellular protein palmitoylation was measured and analyzed via flow cytometry (*Figure 6—figure supplement 1A*). Fluorescent azide labeling of palmitic acid confirmed that total cellular palmitoylation was unchanged between parental and OxR cells (*Figure 6—figure supplement 1B*).

To further examine the relationship between DR4 palmitoylation and TRAIL sensitization in OxR cells, the irreversible palmitoylation inhibitor 2-bromopalmitate (2BP) was used. 2BP is a commonly used palmitate analog that is thought to bind to palmitoyl acyl transferase, forming an inhibitory enzyme complex (*Draper and Smith, 2009*). Treating SW620 OxR cells with 3.5 µM 2BP in combination with TRAIL significantly reduced TRAIL sensitization and increased the IC50 to over 1000 ng/ml (*Figure 6C, E*). 2BP significantly reduced the number of apoptotic cells in both parental and OxR cells, demonstrating the importance of DR4 palmitoylation in TRAIL signaling, particularly in chemoresistant cells (*Figure 6D*). These data suggest a novel mechanism for enhanced DR4/LR colocalization in OxR cells via enhanced DR4 palmitoylation (*Figure 6F*).

## Metastatic CRC patients show sensitivity to TRAIL liposomes despite chemoresistance

Despite promising specificity for cancer cells and low off-target toxicity, TRAIL's translational relevance has been confounded by a short half-life and ineffective delivery modalities (*Stuckey and Shah, 2013*). In recent studies, our lab has demonstrated that TRAIL-coated leukocytes via the administration of liposomal TRAIL can be effective in eradicating circulating tumor cells (CTCs) in the blood of metastatic cancer patients (*Ortiz-Otero et al., 2020*). Briefly, liposomes were synthesized as previously described using a thin-film hydration method, stepwise extrusion to 100 nm in diameter, and decoration with E-selectin and TRAIL via his-tag conjugation (*Mitchell et al., 2014*; *Figure 7A*). Undecorated 'control' liposomes, soluble TRAIL (290 ng/ml; at equivalent concentrations as TRAIL liposomes), and oxaliplatin (at peak plasma concentrations of 5 µM) were used as controls. Blood was collected from 13 metastatic CRC patients who had previously undergone or were currently undergoing an oxaliplatin chemotherapy regimen (*Table 1*). Of these, five patients were analyzed at 2–3 time points over their respective treatment regimens, representing 21 total samples. Blood samples were treated with TRAIL liposomes or control treatments under hematogenous circulatory shear conditions in a cone-and-plate viscometer. TRAIL liposomes significantly decreased the average percentage of viable CTCs in patient blood to 43% compared to just 86% when treated with oxaliplatin (*Figure 7B*). Interpatient variation was dominant in response to TRAIL liposome treatment as the between-patient coefficient of variation was twice as high (CoV = 0.55) as the

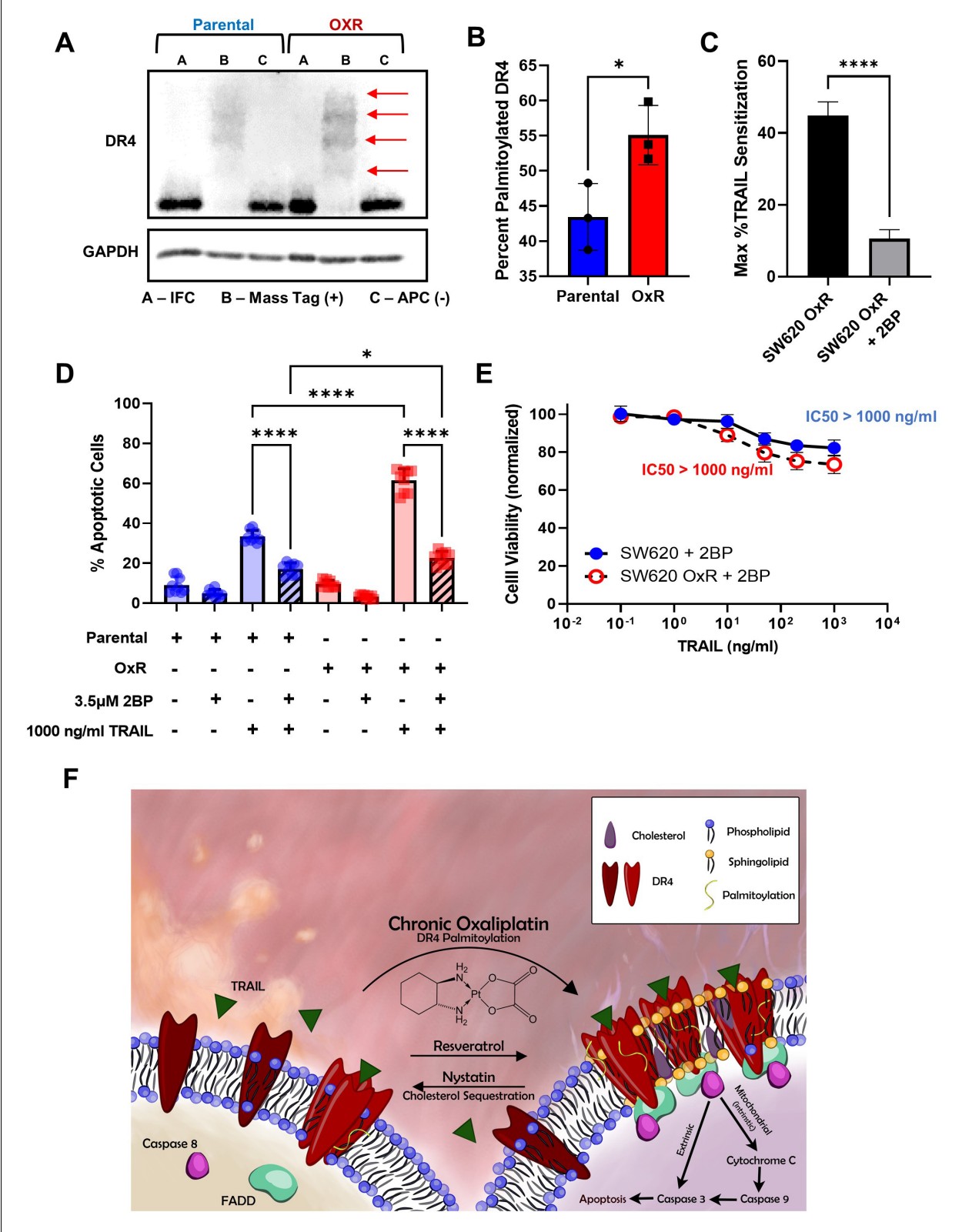

**Figure 6.** Oxaliplatin resistance enhances palmitoylation of DR4, selectively. (**A**) Death receptor palmitoylation was determined by protein precipitation, thioester cleavage, and conjugation of a mass tag to enumerate and quantify the degree of S-palmitoylation between cellular phenotypes. Samples with a mass tag 'B' have distinct bands of equivalent increasing mass, with each mass shift indicating a palmitoylated site. Input fraction control (IFC) samples 'A' were collected before thioester cleavage, while the acyl preservation negative control (APC) samples were incubated with an acyl-

*Figure 6 continued on next page*

*Figure 6 continued*

preservation reagent to block free thiols in place of the mass tag reagent. Arrows show palmitoylation bands. (B) Quantification of the percentage of palmitoylated DR4, calculated by dividing the total palmitoylated mass shift intensity by the average intensity of IFC and APC for each sample. Data are presented as mean ± SD (N = 3). *p<0.05 (unpaired two-tailed t-test). (C) Treatment with the irreversible palmitoylation inhibitor 2BP in combination with TRAIL significantly reduced TRAIL sensitization in SW620 OxR cells. Data are presented as mean + SEM. N = 3 (n = 9). *p<0.0001 (unpaired two-tailed t-test). (D) Percentage of apoptotic SW620 parental and OxR cells after treating with 1000 ng/ml TRAIL and 3.5 µM 2BP in combination (sum of early and late-stage apoptotic cells from Annexin/PI staining). Data are presented as mean + SD. N = 3 (n = 9). *p<0.05; ****p<0.0001 (ordinary one-way ANOVA–Tukey's multiple comparison test). (E) Cell viability determined by Annexin-V/PI staining for cells treated with 0.1–1000 ng/ml TRAIL and 3.5 µM 2BP. IC50 values were calculated using a variable slope four-parameter nonlinear regression. Data are presented as mean ± SD. N = 3 (n = 9). (F) Proposed mechanism of enhanced TRAIL-mediated apoptosis in oxaliplatin-resistant cells.

The online version of this article includes the following source data and figure supplement(s) for figure 6:

**Source data 1.** Quantification of palmitoylated DR4 from western blots (panel B).
**Source data 2.** Cell viability and percent apoptosis in SW620 cells after 2BP and TRAIL combination treatment (panels C-E).
**Source data 3.** Western blot images (raw and annotated) for palmitoylated DR4.
**Figure supplement 1.** Total palmitoylation remains unchanged between SW620 parental and oxaliplatin-resistant (OxR) cells.

average within-patient variation (CoV = 0.28). Viable CTCs were categorized as cells that were cytokeratin(+), DAPI(+), CD45(-), and propidium iodide(-) (*Figure 7C*). TRAIL liposomal therapy reduced total viable CTC counts by 58% compared to control liposomes, and over 32% compared to oxaliplatin after just 4 hr in circulation (*Figure 7—figure supplement 1*). Notably, in two patients (P10 and P11), there were no detectable viable CTCs in blood samples treated with TRAIL liposomes. When categorizing patients by location of metastasis, patients that presented with metastases in the liver or bone showed a greater reduction in viable CTCs (69 and 71%, respectively) than patients with both lung and liver metastases (32%) (*Figure 7D*). Patients had similar CTC reductions regardless of their treatment at the time of blood draw, while those undergoing FOLFOX or capecitabine + oxaliplatin had the highest reduction in CTCs (65 and 60%, respectively) (*Figure 7E*). When categorizing patients as either oxaliplatin-sensitive or -resistant, based on their response to 5 µM oxaliplatin under hematogenous circulatory-shear conditions (threshold 80% CTC viability), there was no significant difference in CTC response to TRAIL liposomes (*Figure 7—figure supplement 2A*). Likewise, grouping patients by those undergoing oxaliplatin chemotherapy and those who had failed oxaliplatin previously, there was no significant difference in reduction of viable CTCs from the administration of liposomal TRAIL (*Figure 7—figure supplement 2B*). This demonstrates the utility of TRAIL liposomes to eradicate CTCs in both oxaliplatin-sensitive and OxR patients.

## CTC DR4-LR colocalization corresponds with TRAIL liposome treatment efficacy and oxaliplatin resistance

Patient CTCs were also stained for DR4 and LRs to examine the relationships between raft colocalization, treatment efficacy, and oxaliplatin resistance. Decreasing LR colocalization with DR4 coincided with reduced efficacy of TRAIL liposomes (higher percentage of viable CTCs after treatment), with a negative slope that significantly deviated from zero (*Figure 7F*). Additionally, increasing LR DR4 corresponded with increasing resistance to oxaliplatin (higher percentage of viable CTCs after oxaliplatin treatment), with a positive slope that significantly deviated from zero (*Figure 7G*). These same trends were observed for total DR4 area (*Figure 7—figure supplement 2C, D*). Despite the small size of the patient cohort, these results are encouraging and support our in vitro data in OxR cell lines. Five patients provided multiple blood samples over the course of their treatment, as shown in *Table 1*. Of these, P07 was the only patient being treated with oxaliplatin (FOLFOX) over the course of all three blood draws. Patient 7 was undergoing the first cycle of FOLFOX at the time of draw 1 and progressed while on FOLFOX for draws 2 and 3. However, DR4 and LR staining of CTCs revealed increased DR4/LR colocalization despite progression (*Figure 7H*). This same trend of enhanced CTC DR4/LR colocalization with treatment was observed in patients undergoing 5FU + Avastin (P01) and FOLFIRI (P09), while P06 (FOLFIRI) exhibited a bimodal response (*Figure 7—figure supplement 2E*). Interestingly, P12 exhibited decreased colocalization in CTCs over the course of treatment. This is hypothesized to be a result of a switch in treatment (FOLFIRI + Avastin to cetuximab + encorafenib) due to progression after the first draw.

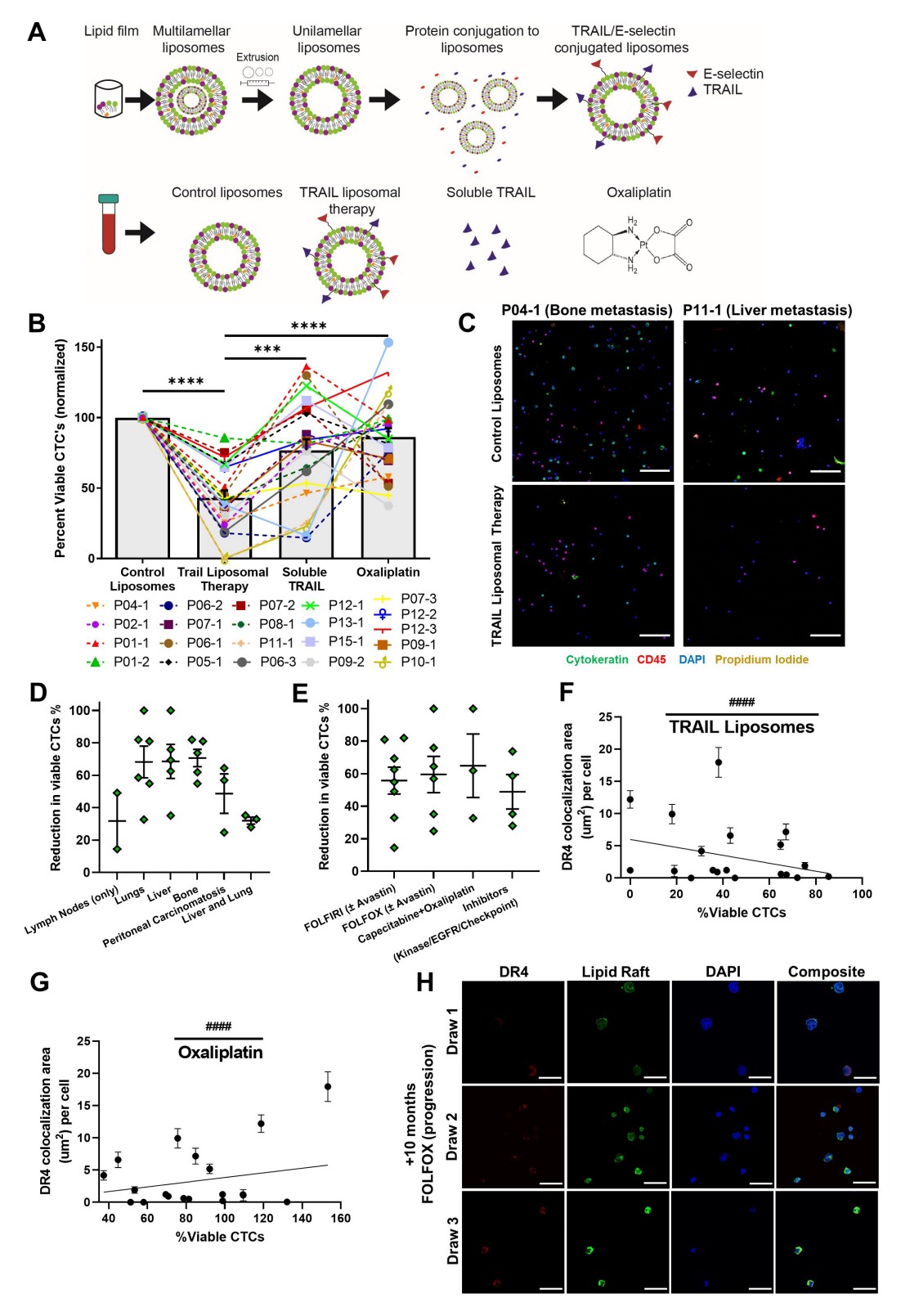

**Figure 7.** TRAIL-conjugated liposomes neutralize circulating tumor cells (CTCs) from the blood of patients with metastatic, oxaliplatin-resistant colorectal cancer. (**A**) Liposomes were synthesized using a thin-film hydration method, followed by extrusion and his-tag conjugation of TRAIL and E-selectin protein. Patient blood samples were treated in a cone-and-plate viscometer under circulatory shear conditions with either control liposomes, TRAIL liposomes, soluble TRAIL, or oxaliplatin. (**B**) Effects of TRAIL liposomes and control treatments on the number of viable CTCs, normalized to

*Figure 7 continued on next page*

*Figure 7 continued*

control liposome treatment. Bars represent the average of all patients and time points (N = 21). **p<0.001; ****p<0.0001 (ordinary one-way ANOVA–Tukey's multiple comparison test). (C) Representative micrographs of two patients showing neutralization of CTCs in TRAIL liposomes compared to control liposomes, stained for cytokeratin (green), DAPI (blue), CD45 (red), and propidium iodide (yellow). Scale bar = 100 μm. (D, E) Reduction in viable CTCs categorized by location of metastasis and treatment administered at the time of blood draw, respectively. (F) DR4/LR colocalization area of patient CTCs plotted against the percentage of viable CTCs following TRAIL liposome treatments. Each point corresponds with one patient draw. ####p<0.0001 (simple linear regression to confirm significant deviation from zero). (G) DR4/LR colocalization area of patient CTCs plotted against the normalized percentage of viable CTCs following oxaliplatin treatment. Each point corresponds with one patient draw. ####p<0.0001 (simple linear regression to confirm significant deviation from zero). (H) CTCs of patient 7, stained for DR4 (red) and lipid rafts (green), demonstrating increased DR4/LR colocalization over the course of 10 months of FOLFOX treatment (with progressive disease despite treatment). Scale bar = 30 μm. For all graphs, data are presented as mean ± SEM.

The online version of this article includes the following source data and figure supplement(s) for figure 7:

**Source data 1.** Viability of patient CTCs following treatment.

**Source data 2.** Correlation analysis of LR-colocalized DR4 area with percent viable CTCs after treatment.

**Figure supplement 1.** Absolute numbers of viable circulating tumor cells (CTCs) per ml of blood following TRAIL liposomal therapy and control treatments.

**Figure supplement 1—source data 1.** Absolute numbers of viable circulating tumor cells (CTCs) per ml of blood following TRAIL liposomal therapy and control treatments.

**Figure supplement 2.** TRAIL liposomes are effective in oxaliplatin-sensitive and -refractory patients.

**Figure supplement 2—source data 1.** Treatment efficacy in oxaliplatin resistant and sensitive patients (panels A and B).

**Figure supplement 2—source data 2.** Correlation analysis of DR4 area in CTCs with percent viable CTCs after treatment.

**Figure supplement 2—source data 3.** Analysis of LR-colocalized DR4 in patient CTCs over time.

**Table 1.** Demographic and clinical information of metastatic colorectal cancer (CRC) patients enrolled in this study.
*Denotes missing treatment analysis for this sample. †Denotes missing DR4/LR analysis for this sample.

| Patient | Age | Sex | Cancer | Metastatic location | Treatment history at draw 1 | Draw 2 | Draw 3 |
|---------|-----|-----|--------|---------------------|-----------------------------|--------|--------|
| P01 | 59 | F | Colon | Paraaortic lymph nodes | †FOLFOX (2016), FOLFIRI, 5-FU + Avastin | +2 months 5-FU + Avastin | * +7 months 5-FU + Avastin |
| P02 | 83 | F | Colon | Liver | †FOLFOX, FOLFIRI, FOLFOX + Avastin | | |
| P04 | 53 | F | Rectal | Pelvis, mesenteric lymph nodes | FOLFOX + Avastin, capecitabine +radiation, regorafenib + nivolumab | | |
| P05 | 68 | M | Rectal | Pulmonary | Capecitabine + oxaliplatin | | |
| P06 | 68 | F | Rectal | Lung, bone | FOLFOX, FOLFIRI | +6 months FOLFIRI | +7 months FOLFIRI |
| P07 | 64 | M | Cecum | Peritoneal carcinomatosis | FOLFOX (1st cycle) | +7 months FOLFOX (progression) | +10 months FOLFOX (progression) Started FOLFIRI |
| P08 | 69 | M | Colon | Lung, abdomen | FOLFOX + Avastin (progression) capecitabine + Avastin, 5-FU + cetuximab + panitumumab | | |
| P09 | 73 | M | Sigmoid | Liver, mesentery | FOLFOX, capecitabine, FOLFIRI, Lonsurf | +7 months FOLFIRI | N/A (patient deceased) |
| P10 | 52 | M | Rectal | Lung | Radiation + capecitabine capecitabine + oxaliplatin | | |
| P11 | 70 | M | Colon | Liver | FOLFOX | | |
| P12 | 59 | M | Colon | Liver, lungs, R adrenal | FOLFOX + Avastin, FOLFIRI + Avastin | Cetuximab + encorafenib (progression) | +3 months cetuximab + encorafenib (progression) |
| P13 | 63 | F | Colon | Adnexa pelvis | FOLFOX, irinotecan + panitumumab, capecitabine + oxaliplatin | | |
| P15 | 79 | M | Colon | Liver | FOLFOX (1st cycle) | | |

## Discussion

Our lab has demonstrated the utility of TRAIL nanoparticles to treat a variety of cancer types in vitro (*Mitchell et al., 2014*), in vivo (*Jyotsana et al., 2019*), and in clinical samples (*Ortiz-Otero et al., 2020*). While front-line chemotherapy remains a viable option for patients with metastatic CRC, long-term treatment frequently leads to chemoresistance, consequently yielding a more aggressive, robust phenotype that is unresponsive to many systemic treatments (*Martinez-Balibrea et al., 2015*). Our results demonstrate that OxR CRC cells are particularly susceptible to TRAIL-mediated apoptosis. Additionally, the ability to eradicate over 57% of OxR CTCs in patient blood demonstrates the utility of TRAIL liposomes clinically. Moreover, two patient samples exhibited 100% neutralization of all viable CTCs following ex vivo TRAIL liposomal treatment. This demonstrates the natural cancer cell targeting ability and low toxicity of TRAIL-based therapeutics, presenting a promising cancer management strategy for patients who have exhausted traditional treatment modalities. TRAIL's apoptotic affect has been shown to be sensitized by circulatory shear stress, further supporting its use as an antimetastatic therapy in the blood of patients (*Mitchell and King, 2013*). Multiple other studies have demonstrated that platin-based chemotherapeutics, including oxaliplatin, are able to sensitize cancer cells to TRAIL-mediated apoptosis when treated in combination (*Cuello et al., 2001*; *Nagane et al., 2000*; *Gibson et al., 2000*; *Toscano et al., 2008*). However, no study has investigated the effects of oxaliplatin resistance on TRAIL-mediated apoptosis, and importantly, no study has demonstrated that OxR cancers can be exploited with TRAIL therapies.

Elucidating the mechanisms that drive OxR TRAIL sensitization will be key in establishing personalized treatment strategies in patients. Interestingly, genetic analysis of TRAIL-sensitized OXR cells demonstrated that OxR cells consistently exhibited downregulated caspase-10. This may seem counterintuitive generally since caspase-10 is a caspase-8 analog that initiates the apoptotic pathway after binding to FADD. However, studies have demonstrated the potential antiapoptotic effects of high caspase-10 expression (*Mühlethaler-Mottet et al., 2011*; *Sprick et al., 2002*). One recent study in particular demonstrated that upon activation with Fas ligand caspase-10 reduced DISC association and activation of caspase-8, rewiring DISC signaling toward the NF-κB pathway and cell survival (*Horn et al., 2017*). However, this noncanonical caspase-10 signaling was found to have an insignificant effect on TRAIL sensitization as evidenced in experiments where caspase-10 was depleted in parental cells. This establishes that the observed augmentation of TRAIL sensitivity is likely a result of a translational or post-translational effect induced by oxaliplatin resistance, rather than a transcriptional change within the apoptotic pathway.

We have demonstrated that augmentation of death receptors, particularly DR4, in OxR cells is one of the drivers of enhanced sensitization. One study found that cisplatin and 5-FU-resistant side populations of colon cancer cells had upregulated DR4, consistent with our results (*Sussman et al., 2007*). While microscopy data suggest that DR5 is upregulated in TRAIL-sensitized OxR cell lines, DR5 area per cell was considerably lower than DR4. Additionally, conflicting western blot and flow cytometry data make the case for DR5 augmentation in OxR cells less convincing. Treatment with the DR4 agonist antibody mapatumumab validated the role of DR4 in the TRAIL sensitization of OxR cells as the differential treatment responses were analogous to that observed from TRAIL treatment. Interestingly, DR4 augmentation appears to be independent from transcriptional upregulation as there was no significant change in mRNA expression between OxR and parental cell lines. Increasing evidence demonstrates that chemoresistance affects small non-coding microRNA (miRNA) expression, which modulates transcriptional and translational processes (*Si et al., 2019*; *Just et al., 2019*). More specifically, studies have shown that oxaliplatin treatment and subsequent resistance in CRC cells alter miRNA expression, affecting signaling pathways within p53, epithelial-to-mesenchymal transition, and cell migration (*Tanaka et al., 2015*; *Gasiulé et al., 2019*; *Evert et al., 2018*). Moving forward, future studies should examine the role of miRNA attenuation post-oxaliplatin resistance on the expression of death receptors, particularly DR4.

While sufficient DR4 expression is important for sustained apoptotic signaling, DR4 localization and compartmentalization within LRs is unequivocally vital. LRs enhance the signaling capacity of surface receptors through a multitude of mechanisms (*Greenlee et al., 2021*). For example, LRs promote death receptor trimerization, which is needed for signal transduction, act as concentrating platforms for DISC assembly and the recruitment of death domains, and protect DRs from internalization or enzymatic degradation (*Simons and Toomre, 2000*). Additionally, juxtaposition of multiple

DR trimers forms supramolecular entities, recently termed 'CASMER' (*Mollinedo and Gajate, 2020*), capable of multivalent TRAIL signaling via extracellular pre-ligand assembly domains (PLADs) (*Naval et al., 2019*). Altering raft integrity via cholesterol sequestration using nystatin had profound impacts on reducing TRAIL sensitization within the OxR phenotype. Moreover, raft stabilization with resveratrol was able to enhance TRAIL sensitization within the parental phenotype, mirroring that observed in OxR cells. These changes in sensitivity were confirmed to coincide specifically with enhanced clustering of DR4 within rafts. These results are consistent with other studies, which have shown that pharmacological alterations of LRs have profound impacts on Fas and TRAIL toxicity (*George and Wu, 2012*; *Delmas et al., 2004*). Other studies have demonstrated that DR4 localization into LRs is obligatory for TRAIL-induced apoptosis in hematological malignancies and non-small cell lung cancer, whereas DR5 has no dependence on raft translocation (*Marconi et al., 2013*; *Naval et al., 2019*; *Ouyang et al., 2013*; *Song et al., 2007*), consistent with our correlative data and receptor contents from LR-isolated membrane fractions. Additionally, one study found that oxaliplatin combination treatment with TRAIL in gastric cancer cells enhances apoptotic signaling through casitas B-lineage lymphoma (CBL) regulation and death receptor redistribution into LRs (*Xu et al., 2009*). While it is evident that rafts promote CASMER formation, death receptor oligomerization, and TRAIL-mediated apoptosis, the mechanism linking the OxR phenotype and enhanced DR4 localization within rafts has yet to be studied.

We have demonstrated that a mechanism for this phenomenon is via enhanced DR4 palmitoylation. Palmitoylation is the post-translational covalent attachment of a fatty acid tail to cysteine residues in the protein transmembrane domain, influencing protein trafficking and signaling. There is evidence that both Fas receptor and DR4 are palmitoylated, while DR5 is not (*Rossin et al., 2009*; *Chakrabandhu et al., 2007*). Furthermore, this post-translational modification has proven to be mandatory for DR4 oligomerization, LR localization, and TRAIL-mediated apoptotic signaling (*Rossin et al., 2009*). Interestingly, in a sensory neuron study in rats, palmitoylation of δ-catenin in dorsal root ganglion was significantly increased after chronic oxaliplatin treatment (*Zhang et al., 2018*). This is analogous to our results as OxR CRC cells that have undergone chronic oxaliplatin treatment exhibited a higher percentage of palmitoylated DR4. Inhibiting palmitoylation with 2BP abrogated the TRAIL-sensitizing effects within OxR cells, demonstrating the mandatory role palmitoylation has on DR4-mediated TRAIL signaling. Additionally, the fact that palmitoylation is inherent to DR4 and not DR5 explains why TRAIL sensitization of OxR cells strongly correlated with LR translocation of DR4, but not DR5. Further studies probing the differences in palmitoylation between parental and OxR phenotypes are warranted to provide a more detailed understanding of oxaliplatin-induced palmitoylation of specific membrane proteins.

We have also shown that these results translate clinically as DR4 expression and LR colocalization of patient CTCs coincided with increased oxaliplatin resistance and increased neutralization of CTCs from TRAIL liposome treatment. Additionally, some metastatic CRC patients exhibited increased DR4/LR colocalization with ongoing chemotherapy cycles despite metastatic progression and worsening prognosis. To our knowledge, this is the first study investigating LR/protein interactions in primary CTCs (*Greenlee et al., 2021*). Overall, our results demonstrate a novel mechanism for TRAIL sensitization in chemoresistant CRC cells via death receptor upregulation and localization within LRs. However, since this sensitization was only observed in two of the four CRC cell lines tested, future studies should investigate genetic and phenotypic differences between these cell lines that may make some more susceptible than others to DR4 palmitoylation, augmentation, and localization. For the scope of this study, we chose to focus on the use of TRAIL treatment alone given its low toxicity and given our previous work in engineering TRAIL-conjugated delivery vehicles. However, since patients are treated with combination therapies, it would be valuable to investigate other therapeutics, such as curcumin or oxaliplatin, that synergize with TRAIL to treat OxR cancer cells (*Ruiz de Porras et al., 2016*; *El Fajoui et al., 2011*). Additionally, future studies should examine the TRAIL sensitization of OxR cells in vivo in orthotopic models of CRC metastasis (*Tseng et al., 2007*). Examining the efficacy of TRAIL and TRAIL-conjugated nanoparticles to curb metastasis of OxR cells in humanized mouse models will provide translational evidence to support the mechanisms elucidated in this study. Moving forward, leveraging the enhanced signaling of death receptors in LRs through mechanisms of drug delivery and LR antagonization will be instrumental in therapeutic development for chemoresistant cancers.

# Materials and methods

## Key resources table

| Reagent type (species) or resource | Designation | Source or reference | Identifiers | Additional information |
|---|---|---|---|---|
| Cell line (*Homo sapiens*) | SW620 adenocarcinoma, colorectal, Dukes' type C | ATCC | #CCL-227 | RRID:CVCL_0547 L15 Media |
| Cell line (*Homo sapiens*) | SW480 adenocarcinoma, colorectal, Dukes' type B | ATCC | #CCL-228 | RRID:CVCL_0546 L15 Media |
| Cell line (*Homo sapiens*) | HT29 adenocarcinoma, colorectal | ATCC | #HTB-38 | RRID:CVCL_0320 McCoy's 5A Media |
| Cell line (*Homo sapiens*) | HCT116 carcinoma, colorectal | ATCC | #CCL-247 | RRID:CVCL_0291 McCoy's 5A Media |
| Cell line (*Homo sapiens*) | SW620 OxR adenocarcinoma, colorectal, Dukes' type C | Kobe Pharmaceutical University | #CCL-227 | RRID:CVCL_4V77 L15 Media |
| Cell line (*Homo sapiens*) | SW480 OxR adenocarcinoma, colorectal, Dukes' type B | MD Anderson Cancer Center Characterized Cell Line Core | #CCL-228 | RRID:CVCL_AU18 L15 Media |
| Cell line (*Homo sapiens*) | HT29 OxR adenocarcinoma, colorectal | MD Anderson Cancer Center Characterized Cell Line Core | #HTB-38 | RRID:CVCL_ 5949 McCoy's 5A Media |
| Cell line (*Homo sapiens*) | HCT116 OxR carcinoma, colorectal | MD Anderson Cancer Center Characterized Cell Line Core | #CCL-247 | RRID:CVCL_4V73 McCoy's 5A Media |
| Chemical compound, drug | Oxaliplatin | MedChemExpress | Cat# HY-17371 | CAS No: 61825-94-3 |
| Commercial assay or kit | MTT Assay Kit | Abcam | Cat# ab211091 | |
| Peptide, recombinant protein | Recombinant Human s TRAIL/Apo2L | PeproTech | Cat# 310-04 | |
| Antibody | Mouse Anti-TNFRSF10A Recombinant Antibody (clone mAY4) | Creative Biolabs | Cat# HPAB-1616-FY | Mapatumumab (0.01–10 μg/ml) |
| Commercial assay or kit | FITC Annexin-V Apoptosis Detection Kit I | BD Pharmingen | Cat# 556547 | Includes propidium iodide |
| Software, algorithm | FlowJo v10.7.1 | FlowJo | RRID:SCR_008520 | |
| Commercial assay or kit | JC1 – Mitochondrial Membrane Potential Assay Kit | Abcam | Cat# ab113850 | |
| Commercial assay or kit | RNeasy Plus Mini Kit | Qiagen | Cat# 74134 | |
| Commercial assay or kit | RT2 First Strand Kit | Qiagen | Cat# 330404 | |
| Commercial assay or kit | RT2 Profiler PCR Human Apoptosis Array | Qiagen | Cat# PAHS-012Z | |

*Continued on next page*

Continued

| Reagent type (species) or resource | Designation | Source or reference | Identifiers | Additional information |
|---|---|---|---|---|
| Software, algorithm | CFX Maestro Software | Bio-Rad | RRID:SCR_018064 | |
| Software, algorithm | GeneGlobe Data Analysis Center | Qiagen | RRID:SCR_021211 | |
| Commercial assay or kit | Gene Knockout Kit v2 – Human CASP10 with Cas9 2NLS Nuclease | Synthego | | sgRNA:Cas9 (90 pmol:10 pmol) |
| Commercial assay or kit | Vybrant Alexa Fluor 488 Lipid Raft Labeling Kit | Invitrogen | Cat# V34403 | |
| Commercial assay or kit | Vybrant Alexa Fluor 555 Lipid Raft Labeling Kit | Invitrogen | Cat# V34404 | |
| Antibody | Mouse anti-human CD261 (DR4) Monoclonal Antibody (clone DJR1) | Invitrogen | Cat# 14-6644-82 | RRID:AB_468188 (1:50 IF) |
| Antibody | Mouse anti-human CD262 (DR5) Monoclonal Antibody (clone DJR2-4) | Invitrogen | Cat# 14-9908-82 | RRID:AB_468592 (1:50 IF) |
| Antibody | Alexa Fluor 555 goat anti-mouse IgG (H+L) | Invitrogen | Cat# A28180 | RRID:AB_2536164 (1:1000 IF) |
| Other | DAPI | Invitrogen | Cat# D1306 | RRID:AB_2629482 (1 µg/ml) |
| Software, algorithm | Fiji – ImageJ | FIJI | | RRID:SCR_002285 JaCOP plugin |
| Antibody | Human TruStain FcX | BioLegend | Cat# 422301 | RRID:AB_2818986 |
| Antibody | PE mouse anti-human CD261 (DR4) (clone DJR1) | BioLegend | Cat# 307206 | RRID:AB_2287472 (5 µl per sample, FC) |
| Antibody | PE mouse anti-human CD262 (DR5) (clone DJR2-4) | BioLegend | Cat# 307406 | RRID:AB_2204926 (5 µl per sample, FC) |
| Antibody | PE mouse anti-human TRAILR3 (DcR1) (clone DJR3) | BioLegend | Cat# 307006 | RRID:AB_2205089 (5 µl per sample, FC) |
| Antibody | PE mouse anti-human TRAILR4 (DcR2) (clone 104918) | BioLegend | Cat# FAB633P | RRID:AB_2205217 (5 µl per sample, FC) |
| Antibody | PE Mouse IgG1 κ Isotype Control (clone MOPC-21) | BioLegend | Cat# 400114 | RRID:AB_326435 (5 µl per sample, FC) |
| Antibody | FITC mouse anti-human DR4 (clone DR-4-02) | Thermo Fisher Scientific | Cat# MA1-19757 | RRID:AB_1955203 (5 µl per sample, FC) |
| Chemical compound, drug | Resveratrol | Sigma-Aldrich | Cat# R5010-100MG | RRID:AB_309682 CAS: 501-36-0 |

*Continued*

| Reagent type (species) or resource | Designation | Source or reference | Identifiers | Additional information |
|---|---|---|---|---|
| Chemical compound, drug | Nystatin | Thermo Fisher Scientific | Cat# BP29495 | CAS: 1400-61-9 |
| Chemical compound, drug | 2-Bromopalmitate | Sigma-Aldrich | Cat# 21604-1G | CAS: 18263-25-7 |
| Antibody | Mouse Anti-Fas Antibody (human, neutralizing) (clone ZB4) | Sigma-Aldrich | Cat# 05-338 | RRID:AB_309682 500 ng/ml (neutralization) |
| Commercial assay or kit | Minute Plasma Membrane-Derived Lipid Raft Isolation Kit | Invent Biotechnologies | Cat# LR-042 | |
| Antibody | DR4 Rabbit monoclonal antibody (clone D9S1R) | Cell Signalling Technologies | Cat# 42533 | RRID:AB_2799223 (1:500 WB) |
| Antibody | DR5 Rabbit polyclonal antibody | Thermo Fisher Scientific | Cat# PA1-957 | RRID:AB_2303474 (1:500 WB) |
| Antibody | Caspase-10 Rabbit polyclonal antibody | Thermo Fisher Scientific | Cat# PA5-29649 | RRID:AB_2547124 (1:1000 WB) |
| Antibody | Mouse anti-human GAPDH (clone 6C5) | MilliporeSigma | Cat# MAB374 | RRID:AB_2107445 (1:2000 WB) |
| Antibody | Mouse Anti-β-Actin monoclonal antibody (clone C4) | Santa Cruz | Cat# sc-47778 | RRID:AB_2714189 (1:1000 WB) |
| Antibody | IRDye 800CW goat anti-rabbit secondary antibody | LICOR | Cat# 926-32211 | RRID:AB_621843 (2:15,000 WB) |
| Antibody | IRDye 800CW goat anti-mouse secondary antibody | LICOR | Cat# 926-32210 | RRID:AB_621842 (2:15,000 WB) |
| Software, algorithm | LICOR housekeeping protein normalization protocol | LICOR Odyssey Fc | RRID:SCR_013715 | |
| Commercial assay or kit | SiteCounter S-Palmitoylated Protein Kit | Badrilla | Cat# K010312 | |
| Commercial assay or kit | EZClick Palmitoylated Protein Assay Kit | BioVision | Cat# K416-100 | |
| Commercial assay or kit | CD45 magnetic beads (human) | Mylteni Biotech | Cat# 130-045-801 | |
| Antibody | Biotin mouse anti-human CD45 Antibody (clone HI30) | BioLegend | Cat# 304004 | RRID:AB_314392 (1:50 IF) |
| Antibody | Streptavidin-conjugated Alexa Fluor 647 | Thermo Fisher Scientific | Cat# S21374 | RRID:AB_2336066 (1:200 IF) |
| Antibody | FITC Mouse Anti-Human Cytokeratin (clone CAM5.2) | BD Pharmingen | Cat# 347653 | (20 µl per sample, IF) |
| Antibody | Goat anti-mouse Alexa Fluor 647 | Thermo Fisher Scientific | Cat# A21235 | RRID:AB_2535804 (1:200 IF) |

## Cell culture

CRC cell lines SW620 (ATCC, #CCL-227), SW480 (ATCC, #CCL-228), HCT116 (ATCC, #CCL-247), and HT29 (ATCC, #HTB-38) were purchased from American Type Culture Collection. SW620 and SW480 cells were cultured in Leibovitz's L-15 cell culture medium (Gibco). HCT116 and HT29 cells were cultured in McCoy's 5A cell culture medium (Gibco). Media was supplemented with 10% (v/v) fetal bovine serum and 1% (v/v) PenStrep, all purchased from Invitrogen. SW480 OxR, HCT116 OxR, and HT29 OxR cells were obtained from MD Anderson Cancer Center Characterized Cell Line Core, supplied and generated by the Dr. Lee Ellis laboratory. SW620 OxR cells were obtained from Dr. Mika Hosokawa at Kobe Pharmaceutical University in Japan. OxR derivative cell lines were cultured in the same medium as their parental counterparts. To prevent phenotypic drift of OxR lines, cells were used within six passages from the time they were received. To prevent chemotherapy-induced cytotoxicity in downstream experiments, oxaliplatin was not supplemented in OxR cell culture media. All cell lines were maintained in a humidified incubation chamber at 37°C and 5% $CO_2$. All cell lines were screened for mycoplasma contamination and tested negative.

## MTT assay

SW620, SW620 OxR, HCT116, HCT116 OxR, HT29, HT29 OxR, SW480, and SW480 OxR cell lines were plated into tissue culture 96-well black-walled plates at a concentration of 3000 cells/well and incubated overnight at 37°C. A 10 mM stock oxaliplatin suspension was created by dissolving oxaliplatin (MedChemExpress) in molecular grade water via sonication and heating. Cell culture media was replaced with oxaliplatin treatments ranging from 0 to 1000 μM and incubated for 72 hr. Following treatments, an MTT assay (Abcam) was carried out according to the manufacturer's protocol. The plates were then read using a plate reader (BioTek μQuant) at 590 nm absorbance using gen5 software. Control wells containing the MTT solution without cells were used for background subtraction.

## Transwell assay

Transwell inserts (6-well with 8 μm pores) (Greiner Bio-one) were evenly coated with 75 μl of a 1 mg/ml collagen solution composed of 3 mg/ml rat tail collagen (Gibco), serum-free media, and 0.2% 1 N NaOH for crosslinking. Inserts were incubated for 20 min at 37°C. After crosslinking, 2.5 ml of 10% FBS media was added to the bottom of the well plate while the top was filled with 1 ml of serum-free media until cells were ready for seeding. Parental and OxR SW480, SW620, HT29, and HCT116 cells were seeded in the collagen-coated inserts at a concentration of 200,000 cells/ml in serum-free media. The transwell inserts were replaced with new serum-free media after 2 days. On day 4, the number of cells that had migrated into the bottom plate was counted using a Thermo Fisher Countess II Automated Cell Counter.

## Annexin-V/PI apoptosis assay

Parental and OxR cell lines were plated at 100,000 cells/well onto 24-well plates and incubated overnight at 37°C. Wells were treated in triplicate with soluble human TRAIL (PeproTech) or treated with the anti-DR4 agonist antibody mapatumumab (Creative Biolabs, clone mAY4) and incubated for 24 hr. All cells were collected by recovering the supernatant and lifting the remaining adhered cells using 0.25% Trypsin-EDTA (Gibco). Cells were washed thoroughly with HBSS with calcium and magnesium (Gibco). Cells were incubated for 15 min with FITC-conjugated Annexin-V and propidium iodide (PI) (BD Pharmingen) at room temperature (RT) in the absence of light and immediately analyzed using a Guava easyCyte 12HT benchtop flow cytometer (MilliporeSigma). Viable cells were identified as being negative for both Annexin-V and PI, early apoptotic cells as positive for Annexin-V only, late apoptotic cells were positive for both Annexin-V and PI, and necrotic cells were positive for PI only. Flow cytometry plots were analyzed using FlowJo v10.7.1 software. Control samples included unstained negative control with no Annexin-V/PI to adjust for background and autofluorescence, and Annexin-V-only and PI-only samples for gating apoptotic and necrotic populations.

The change in cell viability in response to TRAIL treatments between parental and OxR cells for each of the four CRC cell lines was calculated using the following TRAIL Sensitization equation:

$$\text{TRAIL Sensitization} = \frac{(\% \text{Viable Parental Cells}) - (\% \text{Viable OxR Cells})}{(\% \text{Viable Parental Cells})} * 100\%$$

where the percentage of viable cells was normalized to untreated controls for each trial. TRAIL sensitization was calculated for each concentration of TRAIL, where the 'Maximum TRAIL Sensitization' was the highest sensitization observed among all concentrations. Since this sensitization equation is based on a percent reduction formula, small changes in viability can yield large TRAIL sensitizations when cell viability is low. To account for this, both cell viability and TRAIL sensitization are reported to provide a complete perspective on treatment responses between cell lines.

## JC-1 (mitochondrial membrane potential) assay

SW620 and HCT116 cells (parental and OxR) were plated at 100,000 cells/well onto 24-well plates and incubated overnight. Cells were treated in triplicate with TRAIL for 24 hr. Following treatment, cells were collected, washed thoroughly with HBSS without calcium and magnesium, and incubated for 15 min with JC-1 dye (Abcam) in accordance with the manufacturer's protocol. JC-1 fluorescence was assessed via flow cytometry. Cells with healthy mitochondria were identified as having higher red fluorescence while those with depolarized mitochondria had lower red JC-1 fluorescence.

## RT-PCR profiler array

$2 \times 10^6$ SW620 and HCT116 (parental and OxR) cells were seeded into a 100-mm-diameter cell culture dish for 24 hr. Cells were lifted using a cell scraper and washed with HBSS with calcium and magnesium. RNA was isolated using the RNeasy Plus Mini Kit (Qiagen) according to the manufacturer's protocol. RNA yield following isolation was determined using a UV5Nano spectrophotometer (Mettler Toledo). cDNA synthesis was completed using the RT$^2$ First Strand Kit (Qiagen, 330404) using 0.5 µg RNA per sample. RNA expression of 84 apoptotic genes was analyzed using the RT2 Profiler PCR Human Apoptosis Array (Qiagen, PAHS-012Z). Arrays were prepared according to the manufacturer's protocols applied to the prepared cDNA samples. Profiler array plates were run on a CFX96 Touch Real Time PCR (Bio-Rad) using the following protocol: 1 cycle for 10 min at 95°C, 40 cycles of 95°C for 15 s followed by 60°C for 60 s at a rate of 1°C/s. Melt curves were generated immediately following the PCR protocol. Cycle threshold (Ct) values were calculated using CFX Maestro Software (Bio-Rad). Data analysis was completed using the GeneGlobe Data Analysis Center (Qiagen). Volcano plots were generated in GraphPad Prism using calculated fold changes in gene expression between OxR and parental cells and their corresponding p-values.

## CRISPR-Cas9 KO

KO of the CASP10 gene in SW620 cells was completed using the Gene Knockout Kit v2–Human CASP10 kit with Cas9 2NLS Nuclease (Synthego). Ribonucleoprotein (RNP) complexes were made at a 9:1 ratio of sgRNA:Cas9 (90 pmol:10 pmol) in Gene Pulser Electroporation Buffer (Bio-Rad, 1652677) and incubated for 10 min at RT. Cas9 control samples consisted of 10 pmol Cas9 with no sgRNA. RNP complexes were added to 200,000 cells in 200 µl electroporation buffer (0.2 cm cuvette) and electroporated via the Gene Pulser Xcell Electroporation System (Bio-Rad) using exponential decay pulses (145 V, 500 µF, 1000 ohm). Cells were immediately cultured in 12-well plates and allowed to recover for 7 days before measuring KO efficiency.

## Confocal microscopy and image analysis

Parental and OxR cells were seeded onto polystyrene cell culture slides (Thermo Fisher Scientific). Cells were allowed to grow for 48 hr at 37°C. In samples treated with nystatin or resveratrol, cells were plated for 24 hr then treated for 24 hr before staining. Cells were washed and LRs were stained using the Vybrant Alexa Fluor 488 Lipid Raft Labeling Kit (Invitrogen, V34403) according to the manufacturer's protocol. Briefly, cells were incubated with Alexa488-conjugated cholera toxin subunit B (CT-B) followed by an anti-CT-B antibody to crosslink CT-B labeled rafts. Slides were fixed for 15 min with 4% paraformaldehyde (PFA) (Electron Microscopy Sciences) in PBS (Gibco) and then permeabilized using 1% Triton X-100 (MilliporeSigma) in PBS at RT. Slides were blocked for 2 hr with 5% goat serum (Thermo Fisher Scientific) and 5% bovine serum albumin (BSA, Sigma) in HBSS. Primary staining was done overnight at 4°C with either DR4 monoclonal antibody (Invitrogen, clone DJR1) or DR5

monoclonal antibody (Invitrogen, clone DJR2-4) in the blocking serum at a ratio of 1:50. Secondary staining was carried out with Alexa Fluor 555 goat anti-mouse IgG (H+L) (Invitrogen, A28180) for 30 min at RT (1:1000). Slides were stained with DAPI (Invitrogen, D1306) for 30 min at RT in the blocking solution at 1:1000. Washes were done twice between each step for 5 min each using 0.02% Tween20 in PBS. Slides were assembled using 10 µl of Vectrashield antifade mounting media (Vector Laboratories). Confocal imaging was performed using an LSM 880 (Carl Zeiss) with a 63×/1.40 Plan-Apochromat Oil, WD = 0.19 mm objective. At least five images were taken per sample.

Image analysis was performed in FIJI using a macro to automate quantification of raft and DR contents per cell. Briefly, all images were adjusted for background using the same thresholding specifications. The 'analyze particles' feature was used to quantify the total area of LRs and DR per outlined cell. Colocalization events were quantified by creating binary masks of DR and LR events. For each gated cell, the LR and DR binary masks were multiplied to create a binary projection of colocalized events. Raw integrated density and cell area (ROI area) were also measured. Cells with areas outside of three times the standard deviation from the mean were considered outliers and not included in the analysis. Colocalization analysis was also performed using the JACoP plugin in FIJI (*Bolte and Cordelières, 2006*). The MCC was calculated as the fraction of LR colocalized DR4.

### Flow cytometry
#### Surface DR expression
Parental and OxR cell lines were cultured to 70% confluency upon collection and split into 250,000 cells per sample. Cells were fixed in 4% PFA in HBSS for 15 min at RT, then blocked in a 100 µl 1% BSA solution for 30 min at 4°C, with 2× HBSS washes between each step. Cells suspensions of 100 µl were incubated for 15 min at RT with 2 µl Human TruStain FcX (BioLegend, 422301) to prevent nonspecific Fc receptor binding. Samples were immediately stained with 5 µl of either PE anti-human CD261 (DR4) (BioLegend, clone DJR1), PE anti-human CD262 (DR5) (BioLegend, clone DJR2-4), PE anti-human TRAILR3 (DcR1) (BioLegend, clone DJR3), PE anti-human TRAILR4 (DcR2) (R&D Systems, clone 104918), or PE Mouse IgG1 κ Isotype Control (BioLegend, clone MOPC-21) for 30 min at 4°C. Samples were washed twice with HBSS and analyzed using a Guava easyCyte flow cytometer. A chi-squared test was performed using FlowJo v10.7.1, where significance in histogram distribution was confirmed if T(x) between parental and OxR stained samples was greater than T(x) between background (unstained) parental and OxR samples (see *Supplementary file 1*).

#### FRET
Cells were prepared as described above, but without fixation or permeabilization. Samples were stained for LRs using the Vybrant Alexa Fluor 555 Lipid Raft Labeling Kit (Invitrogen, V34404). Samples were then stained with 5 µl FITC anti-human DR4 (Thermo Fisher Scientific, clone DR-4-02) for 30 min at 4°C. Samples were washed twice with HBSS and analyzed using a Guava easyCyte flow cytometer. Donor quenching FRET efficiency was calculated using the following formula:

$$E = 1 - \frac{FI_{LR+DR} - FI_B}{FI_{DR} - FI_B}$$

where E is the FRET efficiency, $FI_{LR+DR}$ is the mean fluorescence intensity of the double-stained LR/DR4 sample (acceptor + donor), $FI_{DR}$ is the mean fluorescence intensity of the DR4-only stain (donor only), and $FI_B$ is the fluorescence intensity of an unstained sample (background). Fluorescence intensity was recorded in the donor (FITC) channel.

### TRAIL combination treatments
#### Resveratrol
Parental SW620 and HCT116 cells were plated at 100,000 cells/well onto 24-well plates and incubated overnight at 37°C. Cells were treated with 70 µM resveratrol (Sigma) in combination with 0.1–1000 ng/ml of TRAIL for 24 hr. Following treatment, cells were collected for Annexin-V/PI apoptosis assay. TRAIL sensitization was calculated using the following equation:

$$TRAIL\,Sensitization_{resveratrol} = \frac{(\%\,Viable\,Parental\,Cells) - (\%\,Viable\,Parental\,Cells_{+resv})}{(\%\,Viable\,Parental\,Cells)} * 100\%$$

where TRAIL + resv treatments were normalized to resveratrol treatment in the absence of TRAIL to account for any resveratrol-associated cytotoxicity.

## Nystatin

SW620 OxR and HCT116 OxR cells were plated at 100,000 cells/well onto 24-well plates and incubated overnight at 37°C. Cells were treated with 5 μM nystatin (Thermo Fisher Scientific) in combination with 0.1–1000 ng/ml of TRAIL. Following treatment, cells were collected for Annexin-V/PI apoptosis assay. TRAIL sensitization was calculated using the following equation:

$$\mathrm{TRAIL\,Sensitization_{nystatin}} = \frac{\left(\%\,\mathrm{Viable\,OxR\,Cells_{+nys}}\right) - \left(\%\,\mathrm{Viable\,OxR\,Cells}\right)}{\left(\%\,\mathrm{Viable\,OxR\,Cells_{+nys}}\right)} * 100\%$$

where TRAIL + nys treatments were normalized to nystatin treatment in the absence of TRAIL to account for any nystatin-associated cytotoxicity.

## 2-Bromopalmitate

SW620 parental and OxR cells were plated at 100,000 cells/well onto 24-well plates and incubated overnight at 37°C. Cells were treated with 3.5 μM 2BP (MilliporeSigma) in combination with 0.1–1000 ng/ml of TRAIL. Following treatment, cells were collected for Annexin-V/PI apoptosis assay. TRAIL sensitization was calculated as described above.

## Anti-Fas (ZB4)

SW620 OxR cells were plated at 100,000 cells/well onto 24-well plates and incubated overnight at 37°C. Cells were treated with 500 ng/ml human anti-Fas (MilliporeSigma, Clone ZB4) with and without 1000 ng/ml of TRAIL. Following treatment, cells were collected for Annexin-V/PI apoptosis assay. TRAIL sensitization was calculated as described above.

## Western blot

LRs were isolated according to the manufacturer's protocol using the Minute Plasma Membrane-Derived Lipid Raft Isolation Kit (Invent Biotech, LR-042). Cell lysates and LR protein isolates were prepared by sonication in 4× Laemmli sample buffer (Bio-Rad, 1610747) and then loaded into 10% SDS-polyacrylamide gels for electrophoresis. Protein transfer onto a PVDF membrane was carried out overnight, and then blocked with Intercept (TBS) Blocking Buffer (LICOR, 927-60001) at RT for an hour. Primary antibody incubation occurred overnight at 4°C for DR4 (Cell Signaling Technology, 42533) and DR5 (Thermo Fisher Scientific, PA1-957) at 1:500 dilution and caspase-10 (Thermo Fisher Scientific, PA5-29649) at a 1:1000 dilution in LICOR buffer. Cell lysate protein bands were normalized to GAPDH (EMD Millipore, MAB347) at 1:2000 dilution, while LR isolates were normalized to β-actin (Santa Cruz, 47778) at 1:1000 dilution in LICOR blocking buffer. Western blots were quantified using the Licor Odyssey Fc with IRDye 800CW goat anti-rabbit secondary antibody (LICOR, 926-32211) and IRDye 800CW goat anti-mouse secondary antibody (LICOR, 926-32210) at a dilution of 2:15,000. Quantification was done following the LICOR housekeeping protein normalization protocol.

## Palmitoylation assay

Cells were grown to 70% confluency in a 100 mm tissue culture dish. Palmitoylation of DR4 was measured using the SiteCounter S-Palmitoylated Protein Kit (Badrilla, K010312) according to the manufacturer's protocol. Input fraction controls (IFC) were obtained prior to thioester cleavage. Acyl preservation negative controls (APC-) were obtained by using an acyl preserving reagent instead of mass-tag conjugation. Western blots were run for DR4 following the 'western blot' protocol described above. The percentage of DR4 palmitoylation was calculated by dividing the total intensity of all palmitoylated bands (mass tag) divided by the average intensity of the IFC and APC(-) bands for that sample.

To measure the amount of total palmitoylated protein, cells were cultured in 96-well plates at a concentration of 20,000 cells/well. The EZClick Palmitoylated Protein Assay Kit (BioVision, K416-100) was used in accordance with the manufacturer's protocol. Cells were incubated overnight with either

1× EZClick Palmitic Acid label in media or culture media with no label (background control). Cells were recovered and stained using EZClick Fluorescent Azide, then analyzed via flow cytometry for shifts in FL2-H intensity. Median fluorescence intensity (MFI) was calculated by subtracting the background intensity from each sample (Palmitic Acid label [-]/ Fluorescent Azide [+]).

## Patient blood samples

Peripheral whole blood samples of 10 ml were collected from 13 metastatic CRC patients after informed consent. Patient criteria for this study included the following: presenting with metastatic CRC at the time of blood draw and undergone (or undergoing) oxaliplatin-containing chemotherapy (i.e., FOLFOX). Additionally, five patients had samples collected through their respective chemotherapy regimens. De-identified blood samples were transported from the Guthrie Clinic to Vanderbilt University and processed within 24 hr. Blood samples were split for treatment (8 ml) and death receptor/LR staining (1–2 ml).

## Ex vivo treatment of CRC patient blood samples

For the treated samples, 2 ml of blood were treated with either 40 µl of control liposomes, 40 µl TRAIL/E-selectin conjugated liposomes (290 ng/ml of TRAIL), 6 µl (290 ng/ml) of soluble TRAIL, or 2 µl (5 µM) of oxaliplatin. Liposomes were synthesized using a thin-film hydration method followed by extrusion and his-tag conjugation as described previously (*Mitchell et al., 2014*). The aliquots were sheared for 4 hr in a cone-and-plate viscometer (Brookfield LVDVII) at a shear rate of 120 $s^{-1}$. Prior to incubation, the cone-and-plate viscometers were blocked using 5% BSA for 30 min. After 4 hr, the blood aliquots were washed from the viscometer's spindle and cup by using twice the volume of HBSS without calcium and magnesium. Blood aliquots were placed over twice the volume of Ficoll (GE Healthcare) to separate out mononuclear cells within the buffy coat. CTCs were enriched using a negative selection kit with CD45 magnetic beads (Mylteni Biotech, 130-045-801) following the manufacturer's protocol (*Ortiz-Otero et al., 2020*).

The resulting isolated CTCs were placed in cell culture overnight using RPMI media supplemented with 10% FBS. After 1 day in culture, the cells were recovered from the tissue culture plate and stained with 100 µl of propidium iodide for 15 min. Cells were washed, fixed with 4% PFA, and cytospun onto microscope slides using a Cytospin 3 (Shandon). Samples were then permeabilized and blocked with 100 µl of 0.25% Triton-X (Sigma) for 15 min and 100 µl of blocking solution (5% BSA and 5% goat serum) for 1 hr, respectively. Cells were stained with anti-CD45 conjugated with biotin (BioLegend, clone HI30) for 45 min at 1:50 dilution. Finally, cells were stained with 100 µl of streptavidin-conjugated Alexa Fluor 647 (Thermo Fisher Scientific, S21374) at 1:200 dilution and 20 µl per sample of anti-cytokeratin conjugated with FITC (BD Pharmingen, clone CAM5.2) for 45 min. Cells were washed 3× after each staining incubation using 200 µl of 0.02% Tween20 in PBS. Cells were stained with 10 µl of DAPI mounting media (Vector Laboratories), covered with a coverslip (no. 1.5, VWR), and sealed with nail polish.

Five images per sample were taken at random locations using an LSM 710 (Carl Zeiss) with a 20×/0.8 objective. The cell number in the sample was scaled up by multiplying by the relative area (slide area/frame area). Viable tumor cells were identified using the following criteria: (1) positive for DAPI, (2) negative for CD45, (3) positive for cytokeratin, and (4) negative for propidium iodide.

## Staining of LR and death receptors in primary CTCs

CTCs from the remainder of the patient blood were isolated and cytospun onto slides as described above. Death receptors and LRs were stained and analyzed as detailed above in 'Confocal microscopy and image analysis.' LRs were stained using the Vybrant Alexa Fluor 555 Lipid Raft Labeling Kit (Invitrogen, V34404) after CTCs were spun onto slides. Secondary staining for DR4 and DR5 was completed using goat anti-mouse Alexa Fluor 647 (Thermo Fisher Scientific, A21235) at a 1:200 dilution. Cells were also stained with FITC-conjugated cytokeratin, as described above, to positively identify CTCs for analysis.

## Statistical analysis

Data sets were plotted and analyzed using GraphPad Prism 9. When comparing two groups, a symmetric unpaired t-test was used with p<0.05 considered significant. One-way ANOVA with multiple

comparisons was used for multiple groups with p<0.05 considered significant. At least three independent biological replicates were used for each experiment unless otherwise stated.

## Acknowledgements

This work was funded by the National Institutes of Health, Grant No. R01CA203991 to MRK. We thank all the cancer patients who donated blood samples for this study. We also thank the Oncology Research Coordinator at Guthrie Clinic, Michelle Hunter, for supervising blood sample collection and shipment. Finally, we thank Dr. Mika Hosokawa and Dr. Lee Ellis for providing us with the oxaliplatin-resistant cell lines, as well as Matthew R Zanotelli for assistance in writing the colocalization macro in FIJI.

## Additional information

### Funding

| Funder | Grant reference number | Author |
|---|---|---|
| National Institutes of Health | R01CA203991 | Michael R King |

The funders had no role in study design, data collection and interpretation, or the decision to submit the work for publication.

### Author contributions

Joshua D Greenlee, Conceptualization, Data curation, Formal analysis, Validation, Investigation, Methodology, Writing - original draft, Writing - review and editing; Maria Lopez-Cavestany, Nerymar Ortiz-Otero, Investigation, Methodology; Kevin Liu, Tejas Subramanian, Investigation; Burt Cagir, Resources; Michael R King, Conceptualization, Supervision, Funding acquisition, Project administration, Writing - review and editing

### Author ORCIDs

Joshua D Greenlee (iD) https://orcid.org/0000-0002-1088-727X
Maria Lopez-Cavestany (iD) https://orcid.org/0000-0001-5358-3746
Burt Cagir (iD) http://orcid.org/0000-0002-9682-3807
Michael R King (iD) https://orcid.org/0000-0002-0223-7808

### Ethics

Human subjects: All of the experiments were done in accordance with the U.S Federal Policy for the Protection of Human Subjects and approved by the Institutional Review Board of the Guthrie Clinic (IRB#1909-42; Approved 10/01/2019). The participants in this study were fully informed regarding the objective of the current study and written consent was obtained.

### Decision letter and Author response

Decision letter https://doi.org/10.7554/eLife.67750.sa1
Author response https://doi.org/10.7554/eLife.67750.sa2

## Additional files

### Supplementary files

- Supplementary file 1. Statistical reporting of chi-squared T(x) values for comparing distribution differences in flow cytometry staining. Significance was determined if chi-squared T(x) sample > chi-squared T(x) background.

- Transparent reporting form

## Data availability

Raw data has been deposited to Dryad, available here: https://doi.org/10.5061/dryad.3xsj3txg3.

The following dataset was generated:

| Author(s) | Year | Dataset title | Dataset URL | Database and Identifier |
|---|---|---|---|---|
| Greenlee JD, Cavestany ML, Ortiz-Otero N, Liu K, Subramanian T, Cagir B, King MR | 2021 | Oxaliplatin Resistance in Colorectal Cancer Enhances TRAIL Sensitivity Via DR4 Upregulation and Lipid Raft Localization | http://dx.doi.org/10.5061/dryad.3xsj3txg3 | Dryad Digital Repository, 10.5061/dryad.3xsj3txg3 |

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
