## [Decision Letter]

**Acceptance summary:**

This manuscript focuses on a potentially new therapeutic vulnerability of colorectal cancer (CRC) cells that are resistant to oxaliplatin treatment. Identifying new therapeutic strategies for metastatic CRC is a critical and urgent need. This study identifies a potentially new therapeutic approach, whereby Oxaliplatin resistant (OxR) cells show increased sensitivity to TRAIL-mediated apoptosis that is mediated by increased palmitoylation of DR4 and localization to lipid rafts. These findings serve as a proof of concept that patients who become resistant to Oxaliplatin therapy may benefit from TRAIL-based therapeutics.

**Decision letter after peer review:**

Thank you for submitting your article "Oxaliplatin Resistance in Colorectal Cancer Enhances TRAIL Sensitivity Via DR4 Upregulation and Lipid Raft Localization" for consideration by *eLife*. Your article has been reviewed by 3 peer reviewers, and the evaluation has been overseen by a Reviewing Editor and Maureen Murphy as the Senior Editor. The reviewers have opted to remain anonymous.

Essential Revisions:

While the findings of the authors are interesting, there are several major concerns with regard to data interpretation and experimental rigor.

1. The authors have not rigorously tested the role of DR4 in the sensitization to TRAIL for example through use of DR4 agonist antibodies used to treat oxaliplatin-resistant cells or through studying the effect of DR4 knockdown on the observed TRAIL sensitization. Similarly, the effect of resveratrol has not been rigorously attributed to DR4. Some data implicates DR5 in the sensitization (Supplementary Figure 6) but in the end the authors emphasize DR4. There could be other mechanisms involved, and this should be addressed.

2. The palmitoylation data in Figure 6 are equivocal. Since DR4 palmitoylation is required for its raft localization and its ability to oligomerize (Ref 35), the manuscript would be greatly enhanced if the authors could perturb palmitoylation and look at the effects on TRAIL sensitization, in order to strengthen their conclusions and render them less correlative.

3. There is very limited data on DR4 expression in CTCs from patient samples. The data included in Figure 7 with regard to DR4 is, to this reviewer, uninterpretable. Furthermore, the role of DR4 expression or its association with oxaliplatin resistance in the clinical samples is very unclear. The histograms looking at DR5 expression in Supplementary Figure 4 do not appear to support the statements about DR5 upregulation in oxaliplatin-resistant cells. A similar experiment is more convincing for DR4 in HCT116 that are oxaliplatin resistant but less convincing for oxaliplatin-resistant SW620 in Figure 3D. The quantifications in Figure 3D do not match the flow data, with oxaliplatin-resistant SW620 appearing to have more significantly increased DR5 than oxaliplatin-resistant HCT116 as comparing to their respective parental lines. These issues need to be thoroughly and rigorously corrected.

4. There is no mechanistic understanding for increased DR4 expression in oxaliplatin-resistant CRC cells. The authors suggest DR4 expression is increased in stem cells and this may be relevant given previous work that implicated a role for c-myc in DR4 induction in CSCs (ref 8). The role of Myc in the resistance to oxaliplatin or the sensitization to TRAIL is not explored here.

5. Can the authors confirm that the OxR cells have increased invasion and motility compared to parental cells? What is their morphology compared to parental cells? Before comparing area of stain positivity per cell (e.g. of DR4, LR, or DR4/LR co-localization), it should be demonstrated that there is no difference in cell size compared to parental cells.

6. To solidify the difference in DR4 expression, the authors should compare the intensity distribution per cell in the confocal microscopy images instead of the area per cell. To determine the area of DR4 per cell, the authors use a threshold for background. To demonstrate that the results are independent of the threshold and the morphological differences between OxR and parental cells, it is necessary to show the intensity distribution (Figure 3 and all following figures). The difference in intensity distribution derived from flow cytometry (Figure 3D) is small. Can the authors confirm significance in deviation using statistical tools (e.g. Kolmogorov-Smirnov test)? The western blot results are encouraging. Can the authors quantify and compare the observed differences from the western blots with statistical significance?

7. Figure 4 shows enhanced co-localization of DR4 and lipid rafts. Similar to the previous comments, the authors use binary masks (which is threshold-dependent) and compare the area per cell, which can be biased by differences in cell size between the two groups. This measure should be normalized by the total cell area. Additionally, the authors should strengthen their statements by using standard methods for co-localization analysis, such as pixel intensity spatial correlation (plugins available with Fiji). In Figure 4A, it appears that the lipid raft staining is not specific to the membrane for SW620-OxR, can the authors comment on this? There should be information on antibody optimization with control tissue in the methods section.

8. There is a lot of variability in the effect of reduced cell viability of circulating tumor cells across patients and draws. Have the authors considered inter- and intra-patient variation when comparing the cell viability with statistical tests?

9. The results in Figure 1 are not firmly supportive of the conclusions. Specifically, 2/4 (50%) of the cell lines tested show no difference in TRAIL sensitivity regardless of Oxaliplatin resistance. This needs to be addressed, and the conclusions toned down. In addition, the claims made on mitochondrial permeabilization are done on only 1 cell line. In the interest of rigor this should be reproduced in at least 2 other cell lines.

10. Volcano plots in Figure 2B show that Fas is dramatically affected in SW620 cells, yet the authors do not pursue this. What is the impact of Fas on sensitivity to TRAIL in this cell line? This is important because these cells show the greatest sensitivity.

11. The authors claim that OxR CRC cells have enhanced co-localization of DR4 into lipid rafts. While this appears to be the case in the imaging in Figure 4A-B, the Western blot data in Figure 4D does not really support this conclusion, as there appear to be extremely modest differences of Lipid Raft DR4 between parental and OxR cells. Can the authors quantify these blots using a western blot approach that uses linear range quantification, such as LiCor, and at the same time, the authors should look at a negative control, such as Lipid Raft DR5 and decoy receptors to show specificity of this result.

---

## [Author Response]

Essential Revisions (for the authors):While the findings of the authors are interesting, there are several major concerns with regard to data interpretation and experimental rigor.1. The authors have not rigorously tested the role of DR4 in the sensitization to TRAIL for example through use of DR4 agonist antibodies used to treat oxaliplatin-resistant cells or through studying the effect of DR4 knockdown on the observed TRAIL sensitization. Similarly, the effect of resveratrol has not been rigorously attributed to DR4. Some data implicates DR5 in the sensitization (Supplementary Figure 6) but in the end the authors emphasize DR4. There could be other mechanisms involved, and this should be addressed.

To further demonstrate the role of DR4 in TRAIL sensitization, we have now treated parental and OxR cells with the DR4 agonist antibody Mapatumumab (Figure 3G-I, Figure 3—figure supplement 7). Mapatumumab is a fully human agonistic monoclonal antibody that has undergone Phase II clinical testing for use in non-small-cell lung cancer patients. OxR cells had a significant increase in the number of apoptotic cells and exhibited decreases in cell viability in a dose-dependent manner that was similar to the effects observed from treating cells with TRAIL. Likewise, the maximum Mapatumumab sensitizations seen in HCT116 and SW620 cells were similar to that observed from treating with TRAIL (Figure 3I).

To investigate the role of DR5 in sensitization from resveratrol/nystatin, we have measured colocalization changes of DR5 from these treatments. We found that resveratrol and nystatin had no significant effects on DR5 lipid raft colocalization, except in SW620 OxR cells where nystatin treatment surprisingly resulted in a slight increase in DR5 colocalization (Figure 5—figure supplement 1).

2. The palmitoylation data in Figure 6 are equivocal. Since DR4 palmitoylation is required for its raft localization and its ability to oligomerize (Ref 35), the manuscript would be greatly enhanced if the authors could perturb palmitoylation and look at the effects on TRAIL sensitization, in order to strengthen their conclusions and render them less correlative.

This is a great suggestion. To provide more evidence for the role of DR4 palmitoylation in sensitization, the well characterized palmitoylation inhibitor 2-bromopalmitate (2BP) was used. Parental and OxR cells were treated in combination with TRAIL and 2BP for 24 hours. 2BP increased the IC50 of SW620 OxR cells from 108 ng/ml to >1000 ng/ml, and significantly reduced the number of apoptotic cells in both parental and OxR cells (Figure 6D-E). The effects of palmitoylation inhibition were most pronounced in OxR cells, and the maximum %TRAIL sensitization was significantly decreased by 2BP treatment (Figure 6C). Treating with the palmitate analogue 2BP has been shown to inhibit DR4 palmitoylation, while DR5 is not palmitoylated (see ref 40 from Rossin et al.,).

3. There is very limited data on DR4 expression in CTCs from patient samples. The data included in Figure 7 with regard to DR4 is, to this reviewer, uninterpretable. Furthermore, the role of DR4 expression or its association with oxaliplatin resistance in the clinical samples is very unclear.

We have provided more data from patient CTCs on the relationships between DR4 expression, DR4/LR colocalization, oxaliplatin resistance, and treatment response from TRAIL liposomes (Figure 7F-G and Figure 7—figure supplement 2C-D). We demonstrate that both DR4 and DR4/LR colocalization correspond with increasing resistance to oxaliplatin (higher normalized percentage of viable CTCs after treating with oxaliplatin) with a positive slope that significantly deviates from zero. Additionally, we show that decreasing DR4 and DR4/LR colocalization coincide with reduced efficacy of TRAIL liposomes (higher percentage of viable CTCs after treatment), with a negative slope that significantly deviates from zero. The intrapatient treatment effects (between blood draws) in relation to DR4 colocalization was moved to the supplement (Figure 7—figure supplement 2E) as well as the effects of oxaliplatin resistance on TRAIL liposome efficacy (Figure 7—figure supplement 2A-B). This was done to ensure we did not overstate the significance of DR4 colocalization with treatment draw, as this relationship was heterogeneous between patients. Likewise, the data showing insignificant changes in treatment response between oxaliplatin sensitive and resistant patients was for a relatively small patient size. Therefore, this was moved to the supplement so that its insignificance is not overstated.

The histograms looking at DR5 expression in Supplementary Figure 4 do not appear to support the statements about DR5 upregulation in oxaliplatin-resistant cells. A similar experiment is more convincing for DR4 in HCT116 that are oxaliplatin resistant but less convincing for oxaliplatin-resistant SW620 in Figure 3D. The quantifications in Figure 3D do not match the flow data, with oxaliplatin-resistant SW620 appearing to have more significantly increased DR5 than oxaliplatin-resistant HCT116 as comparing to their respective parental lines. These issues need to be thoroughly and rigorously corrected.

In accordance with this comment and comment #6, new statistical tests were performed to measure differences in histograms and DR surface expression between parental and OxR cells (see response to comment #6 for more details on the statistical methods used). The flow cytometry histogram data in Supplementary Figure 4E (Now Figure 3—figure supplement 4E) display surface expression of DR5 (for these experiments, cells were not permeabilized as they were in the confocal analysis in panels A-D). That is, the quantification data measuring DR4/DR5 area is from analysis of microscopy images and distinct from the flow cytometry data. Only SW620 OxR cells were found to have significantly increased surface DR5 expression compared to their parental cell lines (See Supplementary File 1). The text within the Results section has been revised to reflect this finding, and to clarify the difference between data showing total death receptor expression and surface expression. We also provide quantifications of western blot data, showing no increase in DR5 expression in HCT116 OxR and SW620 OxR cells. We have provided a possible explanation for this discrepancy, that microscopy data shows that the total receptor area per cell was considerably lower for DR5 compared to DR4. This may result in statistical significance from relatively small changes within receptor content.

In terms of flow cytometry data for surface DR4 expression, we have added that both HCT116 OxR and SW620 OxR cells have significantly increased expression (See Supplementary File 1).

4. There is no mechanistic understanding for increased DR4 expression in oxaliplatin-resistant CRC cells. The authors suggest DR4 expression is increased in stem cells and this may be relevant given previous work that implicated a role for c-myc in DR4 induction in CSCs (ref 8). The role of Myc in the resistance to oxaliplatin or the sensitization to TRAIL is not explored here.

The reviewer raises a good point in suggesting to measure c-Myc expression between parental and OxR cell lines. We measured the changes in c-Myc expression using western blots from three independent experiments and found no increase in c-Myc expression in OxR phenotypes (see Author response image 1).

In fact, SW620 OxR cells actually had a slight decrease in c-Myc expression. Within the Discussion section, we now mention that future studies should examine mechanisms behind this DR4 increase in oxaliplatin-resistant cells. Studies from the labs that derived these cell lines have previously shown that the expression of several microRNAs were significantly changed in OxR phenotypes (see ref 28 Tanaka et al.,). While it may be outside of the scope for this study to examine microRNA expression between cell lines, future studies using these cells could examine these effects in relation to DR4 upregulation. We also emphasize both within the introduction and discussion that changes in DR4 expression alone are often not sufficient to alter TRAIL sensitivity, while lipid raft localization can have much greater effects on apoptosis (see ref 17-20).

5. Can the authors confirm that the OxR cells have increased invasion and motility compared to parental cells? What is their morphology compared to parental cells? Before comparing area of stain positivity per cell (e.g. of DR4, LR, or DR4/LR co-localization), it should be demonstrated that there is no difference in cell size compared to parental cells.

To confirm OxR cells retain their phenotypes in our hands, we have performed Transwell invasion assays to demonstrate the enhanced migration within OxR cells (Figure 1—figure supplement 1B). Studies from the labs that derived these cell lines have demonstrated OxR cells have a notable spindle shape morphology compared to parental cells (see ref 26-28). Despite OxR cells having an increasingly spindle shaped morphology, we calculated that the total cell area remained unchanged between parental and OxR derivatives (Figure 3—figure supplement 1).

6. To solidify the difference in DR4 expression, the authors should compare the intensity distribution per cell in the confocal microscopy images instead of the area per cell. To determine the area of DR4 per cell, the authors use a threshold for background. To demonstrate that the results are independent of the threshold and the morphological differences between OxR and parental cells, it is necessary to show the intensity distribution (Figure 3 and all following figures).

We have demonstrated that parental and OxR cells have no significant changes in cell area (Figure 3—figure supplement 1). We also measured integrated density per cell on images with no thresholding, which yielded very similar results to our DR4 area per cell data (Figure 3—figure supplement 3). That is, HCT116 OxR and SW620 OxR cells had significantly upregulated DR4 whereas minimally sensitized HT29 and SW480 OxR cells did not.

The difference in intensity distribution derived from flow cytometry (Figure 3D) is small. Can the authors confirm significance in deviation using statistical tools (e.g. Kolmogorov-Smirnov test)?

For all death receptor stains using flow cytometry, we added statistics comparing deviations between parental and OxR cells. A chi-squared test was performed and significance in histogram distribution was confirmed if T(x) between parental and OxR stained samples was greater than T(x) between background (unstained) parental and OxR samples. Supplementary File 1 contains T(x) values calculated in FlowJo from chi-squared test analysis. Both HCT116 OxR and SW620 OxR cells were confirmed to have increased surface DR4 expression (Figure 3D). The methods (Flow cytometry: Surface DR expression) were modified to incorporate these analyses.

The western blot results are encouraging. Can the authors quantify and compare the observed differences from the western blots with statistical significance?

For all western blot analyses, we have added quantification and statistics across three independent experiments. While both HCT116 and SW620 OxR cells had increases in DR4 expression from western blots, only SW620 OxR cells were significant from their parental counterparts (Figure 3F).

7. Figure 4 shows enhanced co-localization of DR4 and lipid rafts. Similar to the previous comments, the authors use binary masks (which is threshold-dependent) and compare the area per cell, which can be biased by differences in cell size between the two groups. This measure should be normalized by the total cell area. Additionally, the authors should strengthen their statements by using standard methods for co-localization analysis, such as pixel intensity spatial correlation (plugins available with Fiji).

As we described in responses to comments #5 and #6, we have added analysis that shows parental and OxR cells have no significant changes in cell area between all four cell lines (Figure 3—figure supplement 1). Additionally, we have added supplementary colocalization analysis using the FIJI plugin JACoP (see methods section Confocal Microscopy and Image Analysis). The Manders’ Correlation Coefficient was calculated as the fraction of lipid raft colocalized DR4 (Figure 4—figure supplement 1). HCT116 OxR and SW620 OxR cells were the only cell lines with increased lipid raft colocalized DR4, consistent with our area per cell analysis using binary colocalization projects (Figure 4A-B). To further strengthen our claims of increased DR4/LR colocalization within HCT116 OxR and SW620 OxR cells, we have added new FRET data measured using flow cytometry (See methods section Flow Cytometry: FRET). FRET was calculated using a previously described flow cytometry donor quenching method (see ref 38 from Ujlaky-Nagy et al.). OxR cell lines had significantly increased DR4/LR FRET efficiency compared to their parental counterparts (Figure 4F).

In Figure 4A, it appears that the lipid raft staining is not specific to the membrane for SW620-OxR, can the authors comment on this? There should be information on antibody optimization with control tissue in the methods section.

We have changed the SW620 OxR image shown in Figure 4A to one that is more representative of the data. For lipid raft samples, best results were obtained when staining was completed in accordance with the manufacturer’s protocol, including the recommended antibody concentrations. Additionally, we have added data comparing lipid raft area per cell in parental and OxR cells (Figure 4—figure supplement 3). All cell lines, with exception to the HT29 cells, had insignificant changes in lipid raft quantities.

8. There is a lot of variability in the effect of reduced cell viability of circulating tumor cells across patients and draws. Have the authors considered inter- and intra-patient variation when comparing the cell viability with statistical tests?

We used a mixed effects repeated measures statistical model that separates between-subject variability from within-patient variability (variability between patient draws). Matching was effective, and while the effects of different treatments were significant (p<0.001), the fixed effects of draw number across treatments were insignificant (p=0.37). We found the average intrapatient coefficient of variation was low (0.279) compared to the interpatient coefficient of variation (0.555), and we have added these findings to the Results section. To further demonstrate that the lower intrapatient variation had no significant effect on statistical significance between treatments, we repeated the ANOVA with multiple comparisons analysis after averaging treatment responses for within-patient repeated draws (see Author response image 2).

**Author response image 2. respfig2:** 

These analyses help to establish that the intrapatient variation from repeated draws were not artificially inflating the statistical significance between treatments.

9. The results in Figure 1 are not firmly supportive of the conclusions. Specifically, 2/4 (50%) of the cell lines tested show no difference in TRAIL sensitivity regardless of Oxaliplatin resistance. This needs to be addressed, and the conclusions toned down.

We have added a statement at the end of the discussion that reinforces that these results were only seen in two of the cell lines tested. Consistent with this, we also mention that future studies should examine genetic and phenotypic differences between these cell lines that make some susceptible and others resistant to this mechanism of TRAIL sensitization.

In addition, the claims made on mitochondrial permeabilization are done on only 1 cell line. In the interest of rigor this should be reproduced in at least 2 other cell lines.

We have repeated the JC-1 assay shown for the SW620 cell lines in the parental and OxR HCT116 cells. Similar to the SW620 OxR cells, HCT116 OxR cells had increasingly depolarized mitochondria at high TRAIL concentrations (Figure 1—figure supplement 3). We chose to perform JC-1 assays in these 2 cell lines as they were the only cells that exhibited high TRAIL sensitizations and decreases in IC50 in OxR cells.

10. Volcano plots in Figure 2B show that Fas is dramatically affected in SW620 cells, yet the authors do not pursue this. What is the impact of Fas on sensitivity to TRAIL in this cell line? This is important because these cells show the greatest sensitivity.

This is a good suggestion and an oversight on our part for not including previously. We have added analysis confirming FasR is upregulated in SW620 OxR cells using surface staining and flow cytometry (Figure 2—figure supplement 1A). We then tested the effects of treating SW620 OxR cells with the FasR neutralizing antibody ZB4 at high concentrations (500 ng/ml). FasR neutralization had no significant effects on the number of apoptotic cells or the maximum TRAIL sensitization, confirming that upregulation of Fas was inconsequential on TRAIL sensitivity in SW620 OxR cells (Figure 2—figure supplement 1B-C).

11. The authors claim that OxR CRC cells have enhanced co-localization of DR4 into lipid rafts. While this appears to be the case in the imaging in Figure 4A-B, the Western blot data in Figure 4D does not really support this conclusion, as there appear to be extremely modest differences of Lipid Raft DR4 between parental and OxR cells. Can the authors quantify these blots using a western blot approach that uses linear range quantification, such as LiCor, and at the same time, the authors should look at a negative control, such as Lipid Raft DR5 and decoy receptors to show specificity of this result.

As mentioned in the response to comment #6, we added quantifications for all western blot data. We quantified blots using LiCor software followed by normalization to a β-actin control. We found that both HCT116 OxR and SW620 OxR cells had significantly increased DR4 in lipid raft isolated fractions (Figure 4D-E). We also analyzed DR5 expression in lipid raft isolates as a negative control and found no detectable lipid raft DR5 in either cell line (Figure 4—figure supplement 2D). This is consistent with other studies which have shown that the function of DR4, but not DR5, is reliant on its translocation to lipid rafts (see ref 17, 20, 53 and 54). We have also added corroborating evidence for DR4/LR colocalization using flow cytometry FRET data (see response to comment #7).